# ARHGAP18-ezrin functions as an autoregulatory module for RhoA in the assembly of distinct actin-based structures

Andrew T Lombardo*, Cameron AR Mitchell, Riasat Zaman, David J McDermitt, Anthony Bretscher

Department of Molecular Biology and Genetics, Weill Institute for Cell and Molecular Biology, Cornell University, Ithaca, United States

**Abstract** The location of different actin-based structures is largely regulated by Rho GTPases through specific effectors. We use the apical aspect of epithelial cells as a model system to investigate how RhoA is locally regulated to contribute to two distinct adjacent actin-based structures. Assembly of the non-muscle myosin-2 filaments in the terminal web is dependent on RhoA activity, and assembly of the microvilli also requires active RhoA for phosphorylation and activation of ezrin. We show that the RhoGAP, ARHGAP18, is localized by binding active microvillar ezrin, and this interaction enhances ARHGAP18's RhoGAP activity. We present a model where ezrin-ARHGAP18 acts as a negative autoregulatory module to locally reduce RhoA activity in microvilli. Consistent with this model, loss of ARHGAP18 results in disruption of the distinction between microvilli and the terminal web including aberrant assembly of myosin-2 filaments forming inside microvilli. Thus, ARHGAP18, through its recruitment and activation by ezrin, fine-tunes the local level of RhoA to allow for the appropriate distribution of actin-based structures between the microvilli and terminal web. As RhoGAPs vastly outnumber Rho GTPases, this may represent a general mechanism whereby individual Rho effectors drive specific actin-based structures.

*For correspondence:
ATL73@cornell.edu

Competing interest: The authors declare that no competing interests exist.

## Editor's evaluation

This important study demonstrates that ARHGAP18, through its recruitment and activation by ezrin, fine-tunes the local level of RhoA to allow for the appropriate distribution of actin-based structures between the microvilli and terminal web. The data use state-of-the-art imaging methods and strongly support the conclusions.

## Introduction

Nearly all vertebrate cells are polarized with morphologically distinct plasma membrane regions supported by cytoskeletal filaments. With a few exceptions, the morphology of the cell cortex is determined by actin cytoskeleton-based structures. Regulation of these structures is controlled largely through active Rho family GTPases that signal to a multitude of effector proteins. The activation of Rho family proteins is mediated by guanine nucleotide exchange factors (GEFs) and inactivation by GTPase-Activating Proteins (GAPs). How Rho signaling is limited to a specific location, in the appropriate abundance, at the correct time has been an area of intense interest (reviewed in *Denk-Lobnig and Martin, 2019*).

Mammalian epithelial cells of the gut, kidney, and placenta are characterized by a polarized morphology where microvilli form at the apical cortex and are excluded from forming on other membrane surfaces. Each microvillus consists of an ~1-μm-long rod-like extrusion of the plasma membrane supported by a core bundle of actin filaments. At the base of the microvilli, and immediately beneath the apical plasma membrane surface, is a non-muscle myosin-2 rich terminal web of interwoven actin (*Mooseker et al., 1978*). Both the attachment of the plasma membrane to the microvillar core bundle and the activation of myosin-2 in the terminal web require active RhoA but through distinct effectors. How these two adjacent actin-based arrangements are assembled and distinctly

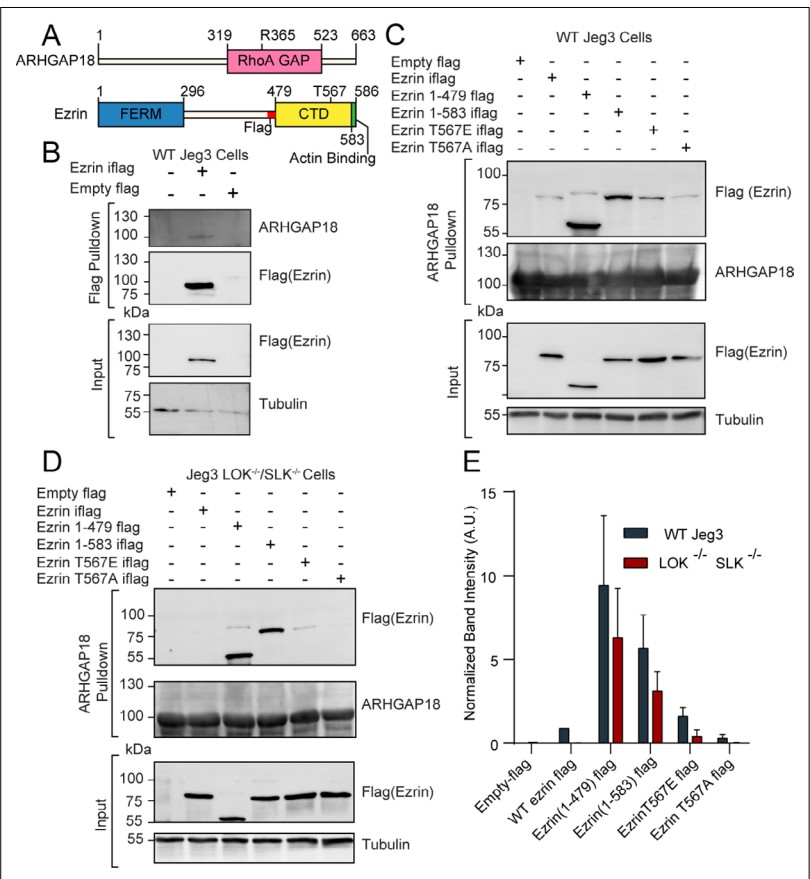

**Figure 1.** ARHGAP18 binds active, open ezrin through the FERM domain. (**A**) Protein domain schematics of human ARHGAP18 and ezrin. (**B**) Western blot from pulldown experiment from passing WT-Jeg3 placental epithelial cell lysate over an anti-flag resin then blotting against flag-tagged ezrin and endogenous ARHGAP18. Ezrin tagged internally (iflag) within a flexible linker region at residue 479. (**C**) Western blot of ARHGAP18-bound resin pulldown experiment blotting against ezrin constructs expressed in WT-Jeg3 cells. (**D**) Western blot of ARHGAP18-bound resin pulldown experiment blotting against ezrin-constructs expressed in LOK$^{-/-}$SLK$^{-/-}$-Jeg3 cells. (**E**) Quantification of normalized band intensity from all experiment presented in (**C, D**). Bars represents mean ± SEM; n = 4. Aggregated WT and LOK$^{-/-}$SLK$^{-/-}$ conditions as populations were compared using a ratio paired *t*-test and found to be significantly different (p=0.0235).

The online version of this article includes the following source data and figure supplement(s) for figure 1:

**Source data 1.** Full western blot images from all blots shown in *Figure 1*.

**Figure supplement 1.** ARHGAP18 affinity for ezrin using purified proteins.

**Figure supplement 2.** ARHGAP18 binds active, open ezrin through the FERM domain in DLD-1 cells.

**Figure supplement 1—source data 1.** Full western blot images from all blots shown in *Figure 1—figure supplement 1*.

**Figure supplement 2—source data 1.** Full western blot images from all blots shown in *Figure 1—figure supplement 2*.

regulated through RhoA is not understood. In this study, we shed light on how this is achieved through the recruitment and activation of a specific RhoGAP to microvilli.

The attachment of the plasma membrane to the underlying actin bundle in microvilli depends on members of the ezrin, radixin, and moesin (ERM) family of proteins (*Bretscher, 1983*; *Casaletto et al., 2011*; *Fehon et al., 2010*). The human ERMs share a similar domain structure with an N-terminal membrane binding band 4.1 ERM (FERM) domain linked through an α-helical region to a C-terminal ~80 residue domain that binds tenaciously to the FERM domain to mask membrane protein binding sites on the FERM domain and an actin-binding site in the C-terminal domain (*Gary and Bretscher, 1995*; *Turunen et al., 1994*; *Figure 1*). ERMs interact with the apical surface of cells by binding of the FERM domain to phosphatidylinositol 4,5-bisphosphate (PI(4,5)P$_2$) (*Fievet et al., 2004*). Once there, the closely related kinases LOK or SLK are specifically targeted to microvilli where they phosphorylate a conserved ERM threonine in the C-terminal domain (T567 in ezrin) (*Belkina et al., 2009*; *Pelaseyed et al., 2017*). Phosphorylation opens ERMs exposing the binding sites for F-actin and for the plasma membrane proteins (*Matsui et al., 1998*; *Nakamura et al., 1995*; *Turunen et al., 1994*). In the absence of this phosphorylation of ERMs, microvilli do not form in human cells (*Zaman et al., 2021*). The actin bundle within microvilli is anchored into a terminal web of actin filaments and non-muscle myosin-2, where force, tension, and, microvilli length are regulated in a carefully balanced system (*Chinowsky et al., 2020*; *Zaman et al., 2021*). A proposed model suggests that non-muscle myosin-2 regulates microvillus length by depolymerizing the actin core bundle where it embeds into the terminal actin web (*Meenderink et al., 2019*).

Activation of non-muscle myosin-2 occurs via phosphorylation of the regulatory light chains, allowing for assembly into bipolar filaments of 14–30 individual myosin molecules (*Sellers and Heissler, 2019*). Phosphorylation occurs predominantly through the activity of the RhoA effector Rho-associated Kinase (ROCK) (*Heissler and Manstein, 2013*). A signaling connection between Rho-GTPases and ERM activation was first suggested in work using rodent model systems (*Tsukita et al., 1994*). RhoA was subsequently reported to act upstream of ERM phosphorylation by the ability of ROCK to directly phosphorylate radixin, although this was subsequently found not to be physiologically relevant (*Hirao et al., 1996*; *Matsui et al., 1998*). A clearer connection was expanded by a report in the fly that moesin acts antagonistically to RhoA (*Speck et al., 2003*). However, unraveling the connections between RhoA signaling and ERMs in humans has been complicated by two factors: first, a controversy relating to the relevant endogenous ERM kinase (see *Sauvanet et al., 2015* for review), which are now known to be LOK and SLK (*Machicoane et al., 2014*; *Viswanatha et al., 2012*; *Zaman et al., 2021*). Second, the presence of genetic redundancy in humans. Humans have three genes specifying the ERM proteins, 2 specifying activating kinases (LOK and SLK), 20 Rho family of GTPases, and at least 49 Rho GTPase activating proteins (ARHGAPs). The functional redundancy of ERMs is most clearly seen by contrasting the relatively mild defects found in moesin knockout mice (*Doi et al., 1999*), with the removal of the single ERM homolog moesin in *Drosophila*, which is lethal (*Speck et al., 2003*).

In humans, both the Rho-GTPase family and ERMs are direct participants in cell proliferation and cancer progression and thus have generated significant clinical attention (*Hoskin et al., 2019*; *Martin et al., 2003*). Despite this importance, the molecular details between these signaling pathways at the plasma membrane have remained elusive. However, a recent report showed that SLK is activated by RhoA (*Bagci et al., 2020*). In a concurrent work, we confirmed that RhoA activates LOK or SLK and further showed that these kinases are required for microvilli formation through their ability to phosphorylate ERM proteins. Additionally, we discovered that in the absence of this system the level of active RhoA is enhanced, although the mechanism was not understood (*Zaman et al., 2021*). Two reports have suggested a possible involvement of Rho GTPase-Activating Protein 18, also called ARHGAP18, in regulating ERM proteins. First, in a proteomic screen in human cells, ARGHAP18 was identified as a candidate interactor of active ezrin (*Viswanatha et al., 2013*). Second, a two-hybrid study in the fly identified a potential homolog of ARHGAP18, known as Conundrum, by its ability to bind the FERM domain of active moesin and to fly RhoA. Further, loss of Conundrum affected epithelial morphogenesis, but genetic experiments were unable to define a specific role in moesin regulation (*Neisch et al., 2013*). Additionally, ARHGAP18 has been shown biochemically to be a specific GAP for RhoA (*Maeda et al., 2011*). In this report, we integrate these findings by showing how ARHGAP18, through its recruitment and activation by microvillar ezrin, forms an autoregulatory module to fine-tune

the local level of active RhoA. This local regulation of RhoA allows for the appropriate distribution of actin-based structures between the microvilli and terminal web. This type of regulatory system is likely to be generally applicable given the large number of Rho GTPases.

## Results

### ARHGAP18 binds active, open ezrin through its FERM domain

ARHGAP18 is a specific RhoA GTPase-Activating Protein (GAP) (*Maeda et al., 2011*) that was also identified as a potential ezrin interactor in a proteomic screen by our group (*Viswanatha et al., 2013*). To characterize the proteomics finding, we expressed a full-length flag-tagged ezrin in the human placental epithelial cell line Jeg3 and performed an anti-flag pulldown experiment (*Figure 1A and B*). Endogenous ARHGAP18 co-precipitated with flag-ezrin (*Figure 1B*). To explore this interaction further, cell lysates from Jeg3 cells expressing various flag-tagged ezrin constructs were passed over a resin to which bacterially expressed recombinant human-ARHGAP18 was bound. Truncated ezrin 1–479 lacking the C-terminal regulatory domain bound at approximately tenfold the level compared to the full-length protein (*Figure 1C and E*). These results indicate that ARHGAP18 binds to a region of ezrin within the FERM domain that is unavailable in the full-length closed protein.

To determine whether the interaction between the FERM domain of ezrin and ARHGAP18 was direct, we purified both proteins using bacterial recombinant expression (*Figure 1—figure supplement 1A*). Either purified FERM or full-length ezrin was passed over a SUMO-ARHGAP18 column and the bound and flow-through fractions were analyzed (*Figure 1—figure supplement 1A*). Purified ezrin FERM domain bound preferentially to the SUMO-ARHGAP18 column compared to full-length ezrin. Using isothermal calorimetry (ITC), the in vitro binding affinity of purified ARHGAP18 to full-length ezrin or the FERM domain was determined (*Figure 1—figure supplement 1B*). The FERM bound with a dissociation constant of $K_d$ = 19.29 ± 3.53 µM (mean ± SEM) while binding affinity was below measurable detection by ITC from the full-length ezrin (*Figure 1—figure supplement 1B*). Therefore, ezrin's C-terminal region inhibits the FERM domain from binding ARHGAP18. To test whether activation and opening of full-length ezrin enhanced ARHGAP18's ability to access the FERM domain, both phosphomimetic ezrin-T567E and constitutively open ezrin (1–583) mutants were expressed in Jeg3 cells (*Figure 1C*). Ezrin's T567 is the target of the activating kinases LOK and SLK, while the ezrin (1–583) construct is a four amino acid truncation previously shown to disrupt the interaction between ezrin's C-terminal and FERM domains (*Viswanatha et al., 2012*; *Zaman et al., 2021*). Both these mutations greatly enhanced ezrin's ability to bind ARHGAP18 compared to the wildtype-ezrin or the non-phosphorylatable ezrinT567A mutant (*Figure 1C and E*). Previous studies have shown that the ezrin-T567 phosphomimetic construct does not achieve a fully open state (*Pelaseyed et al., 2017*; *Viswanatha et al., 2013*; *Zaman et al., 2021*). This is reflected in the western blot's lower binding signal compared to the ezrin (1–583) construct which is fully open.

Phosphorylation at ezrin-T567 is abolished in the absence of the kinases LOK and SLK (*Zaman et al., 2021*). Strikingly, replication of the experiment in cells genetically lacking LOK and SLK showed no binding of full-length wildtype ezrin to ARHGAP18 (*Figure 1D and E*). However, phosphomimetics or truncation-activated ezrin constructs were still capable of binding ARHGAP18, albeit at a somewhat diminished level (*Figure 1E*). Using the DLD-1 colorectal cell line, the interaction of ezrin and ARHGAP18 was similarly phosphorylation site-dependent (*Figure 1—figure supplement 2*), indicating that this interaction was not limited to Jeg3 cells. We conclude that ARHGAP18 interacts with ezrin directly through a region on the FERM domain masked in inactive ezrin but available when phosphorylated at T567 by LOK or SLK.

### ARHGAP18 localizes to microvilli and reduces ezrin T567 phosphorylation to regulate apical microvilli organization and dynamics

We next looked at the localization of ARHGAP18-flag in fixed Jeg3 cells using confocal microscopy. Cells expressing the construct had reduced numbers of microvilli depending on the expression level compared to wildtype Jeg3 cells (*Figure 2A*). Additionally, the ARHGAP18-flag localized with actin and ezrin within the apical microvilli of the cells (*Figure 2A*, inset). We were interested in whether ARHGAP18 localized to just the base of microvilli where its downstream target, RhoA, is believed to be active (*Shaw et al., 1998*). Increased resolution afforded by structured illumination microscopy

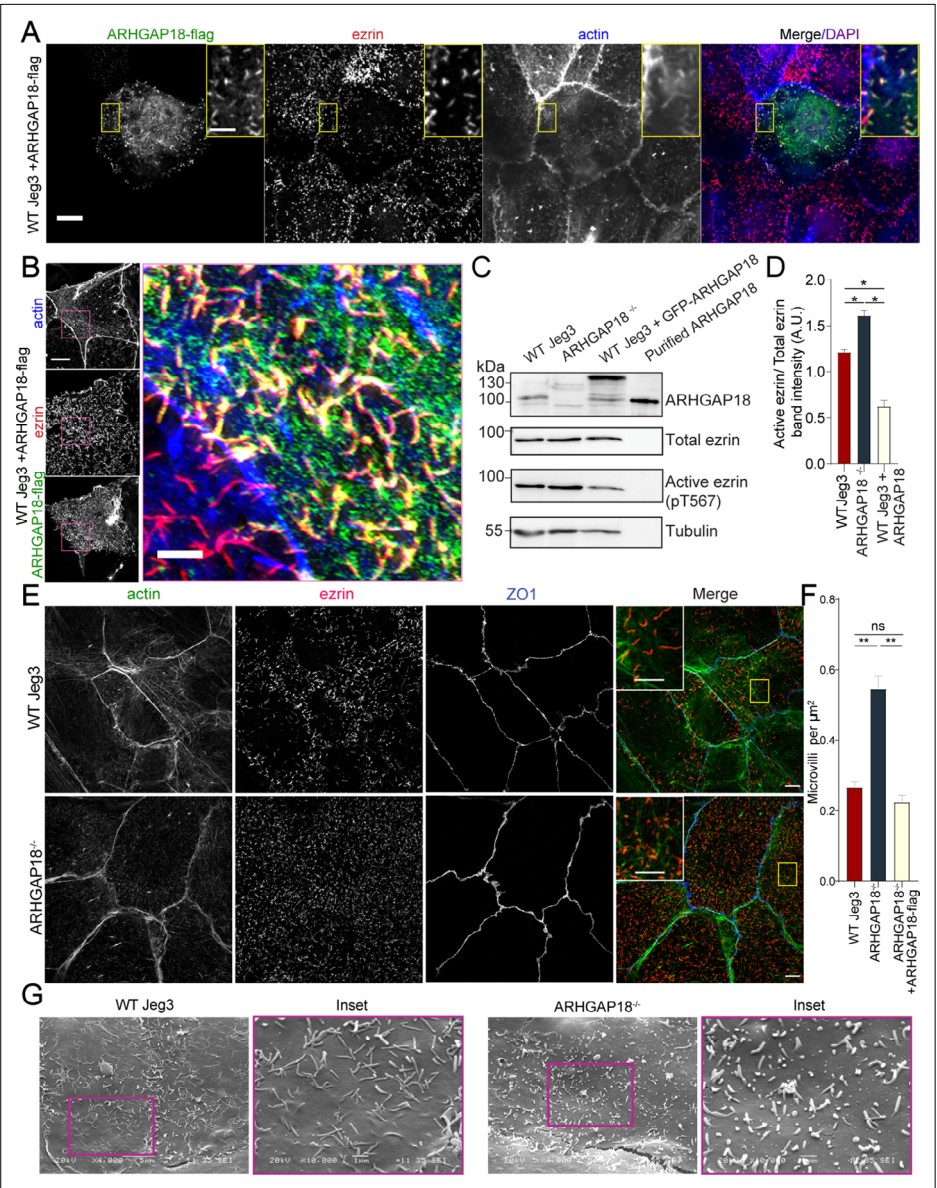

**Figure 2.** ARHGAP18 localizes to microvilli and alters ezrin T567 phosphorylation. (**A**) Confocal immunofluorescence images of WT-Jeg3 cells expressing ARHGAP18-flag. Scale bar 10 µm; yellow box inset scale 2 µm. (**B**) Structured illumination microscopy (SIM) reconstructions of WT-Jeg3 cells expressing ARHGAP18-flag with zoom in on boxed area. Scale bar 10 µm; inset 2 µm. (**C**) Western blot showing CRISPR KO of endogenous ARHGAP18 and ezrin pT567 phosphorylation relative to total ezrin. Antibody against ARHGAP18 was confirmed in part by identification of endogenous ARHGAP18, GFP-ARHGAP18, and purified ARHGAP18 while showing no detection in KO cells. (**D**) Quantification of normalized band intensity from representative experiments presented in (**C**). Bars represent mean ± SEM; n = 4 paired *t*-test; *p≤0.05. (**E**) SIM images of WT vs. ARHGAP18$^{-/-}$ Jeg3 cells with ezrin as a marker for the abundance of microvilli at the apical surface. Scale bar 10 µm; yellow box inset scale 2 µm (**F**) Quantification of the density of microvilli on the surface of Jeg3 cells in WT, ARHGAP18$^{-/-}$, and ARHGAP18 overexpression conditions. Microvilli counts per cell were normalized to the area of each cell to account for cell size variability. Bars represents mean ± SEM; n = 3 replicates; *t*-test; **p≤0.001. (**G**) Scanning electron microscopy (SEM) micrographs of WT-Jeg3 cells' apical surface compared to those of ARHGAP18$^{-/-}$ cells. Left panels imaged at 4000×; scale 5 µm. Right panels imaged at 10,000× within of the boxed areas; scale 1 µm.

The online version of this article includes the following source data for figure 2:

**Source data 1.** Full western blot images from all blots shown in *Figure 2*.

(SIM) revealed that the ARHGAP18 signal was throughout the microvilli, overlapping with the ezrin signal (*Figure 2B*). These data suggest that ARHGAP18 was targeted to microvilli through its ability to bind active ezrin.

To investigate the effect of the loss of ARHGAP18 on the presence of microvilli and ezrin regulation, CRISPR/CAS9 was used to genetically knock out endogenous ARHGAP18 within Jeg3 cells (*Figure 2C and D*). Imaging of ezrin and the quantification of microvilli density in the ARHGAP18$^{-/-}$ cells revealed an increase in the number of microvilli compared to wildtype cells (*Figure 2E and F*). Scanning electron microscopy (SEM) of the apical surface of the cells (*Figure 2G*) showed numerous smaller but variable-sized microvilli in the ARHGAP18$^{-/-}$ cells compared to the wildtype Jeg3 cells. Since phosphocycling of ERMs at the conserved ezrin T567 site is required for microvilli assembly and maintenance (*Viswanatha et al., 2012*), western blotting was performed to test whether loss of ARHGAP18 influenced levels of ezrin phosphorylation. Knocking out ARHGAP18 resulted in an approximately 30% increase in the level of ezrin T567 phosphorylation, while overexpressing ARHGAP18 resulted in the opposite effect (*Figure 2C and D*). It should be noted that these are significant effects given that, in unperturbed cells, the steady-state level of phosphorylation is about 50% of total cellular ezrin (*Viswanatha et al., 2012*).

To see what effect the enhanced level of phospho-ezrin has on microvillar dynamics, we transfected wildtype and ARHGAP18$^{-/-}$ cells with GFP-EBP50, a construct we have used for its excellent live-cell imaging of microvilli, where we reported an average turnover rate of microvilli on the 7–15 min timescale in WT Jeg-3 cells (*Garbett and Bretscher, 2012*). In the ARHGAP18$^{-/-}$ cells, we observed not only the expected increased numbers of shorter microvilli but also increased turnover of individual microvilli in the knockout cells compared to wildtype cells (*Figure 3A*, *Video 1*). To quantify this observation, we tracked the movement and turnover of hundreds of individual microvilli over a 10 min timeframe (*Videos 2 and 3*). Tracking of the individual microvilli showed that in contrast to microvilli in wildtype cells, those in the ARHGAP18$^{-/-}$ cells present at the beginning of a time series (*Figure 3A*, blue arrows) regularly disappeared and were replaced by new microvilli (*Figure 3A*, magenta arrows) over a period of 10 min. Across all the tracked microvilli, we measured the average minimum lifetime for microvilli to be 9.8 ± 2.5 min in wildtype vs. 5.6 ± 3.2 min in ARHGAP18$^{-/-}$ cells (*Figure 3B*). In both conditions, microvilli moved across the apical surface with an overall bias for retrograde trajectories from the apical periphery toward the cell center (*Videos 2 and 3*). We conclude that ARHGAP18 is involved in a negative feedback loop impacting ezrin phosphorylation that regulates microvilli abundance, morphology, and dynamics.

## ARHGAP18 function is dependent on ezrin binding and GAP activity

An attractive model is that ARHGAP18's binding to ezrin targets ARHGAP18 to locally reduce RhoA activity within microvilli. To investigate this model, we expressed various ARHGAP18-flag constructs in the ARHGAP18$^{-/-}$ cells and explored how well they could function to restore normal microvillar organization (*Figure 3C*). Expression of full-length tagged ARHGAP18 rescued the increased microvilli phenotype back to wildtype levels (*Figures 2F and 3D*). On the other hand, expression of a tagged, truncated ARHGAP18 with only the structurally predicted GAP domain (GAP-flag), was more diffuse in its cellular localization and lacked the enrichment to the apical surface as seen with expression of the wildtype protein (*Figure 3D and F*, side projections). Additionally, ezrin targeting and microvilli formation were severely disrupted in the GAP-flag condition (*Figure 3E*). Expression of full-length ARHGAP18 with a point mutation ablating the catalytic 'arginine finger' (ARHGAP18(R356A)-flag) (*Figure 3B*) was targeted correctly to microvilli (*Figure 3E*, inset). However, it had no observable influence on the increased microvilli phenotype characteristic of ARHGAP18$^{-/-}$ cells, indicating that GAP activity is required for rescue (*Figure 3F*). These results suggest that ARHGAP18 must be correctly targeted to active ezrin in microvilli to mediate RhoA regulation for the cell to maintain normal microvilli. To test this model, we expressed a chimeric protein with ARHGAP18's GAP domain fused to the 37 amino acid ezrin binding motif of EBP50 (GAP-flag-EBP50(tail)) (*Figure 3C and E*). This construct was targeted to microvilli (*Figure 3E*, inset) and reduced the number of microvilli on ARHGAP18$^{-/-}$ cells closer to wildtype levels (*Figures 2F and 3F*). To define the specific amino acid region of ARHGAP18 responsible for ezrin interactions, we utilized the COSMIC[2] platform (*Cianfrocco et al., 2017*) to predict the structural interaction via AlphaFold2 (*Jumper et al., 2021*). Utilizing our ARHGAP18 to ezrin interaction pulldown results (*Figure 1C*), we constrained the interaction prediction to the FERM

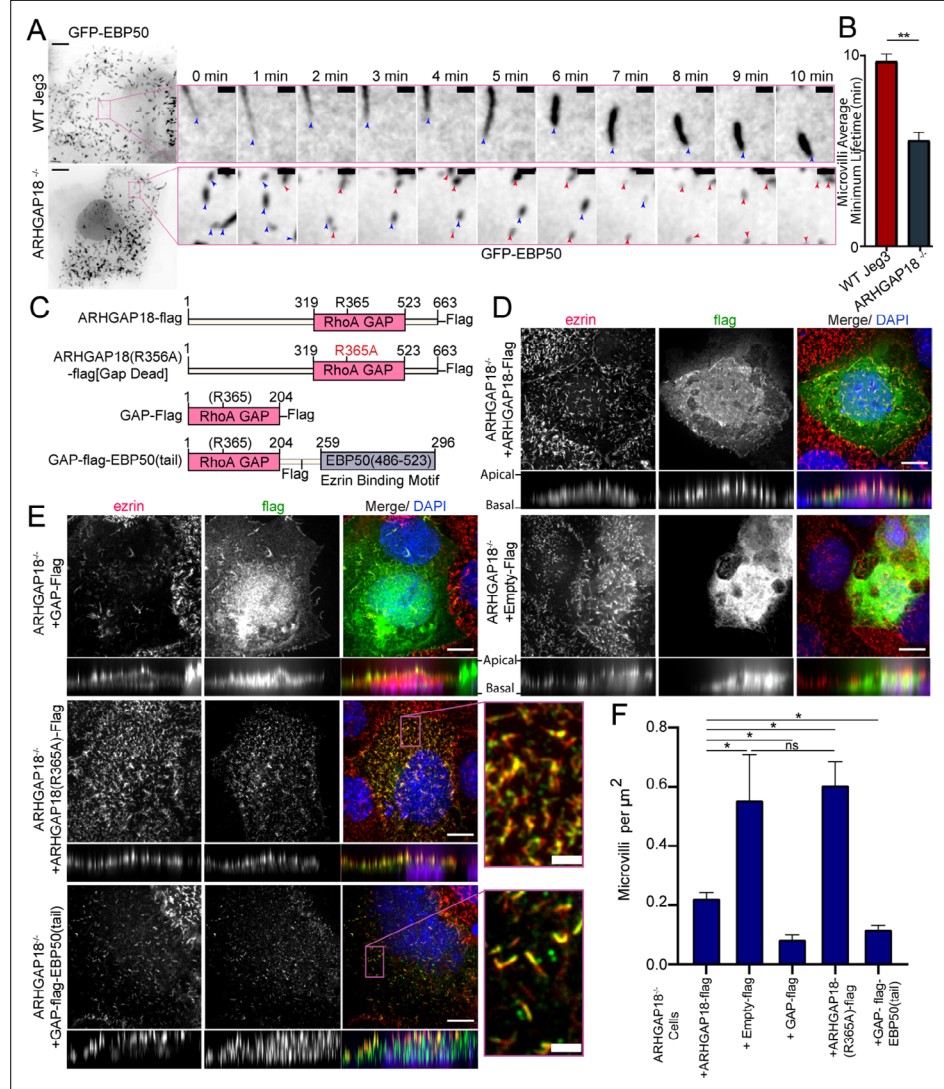

**Figure 3.** ARHGAP18 KO rescue is dependent on ezrin binding and GAP-activity. (**A**) Larger images are the first frame of *Video 3* with inset zoom-in of the magenta box showing frames from the time course. Each frame shows the tip of the microvilli present in the inset image at *t* = 0 (blue arrowhead). In the ARHGAP18$^{-/-}$ condition, the original microvilli disappear and are replaced by new microvilli not present in the first image (magenta arrowheads). Scale bars 10 µm; inset 1 µm. (**B**) Quantification of minimum microvilli lifetime from (**A**) (bars represents mean ± SEM; n = 151; *t*-test; ** p≤0.001). (**C**) Schematic of constructs used in this figure. (**D**) Immunofluorescence confocal images of ARHGAP18$^{-/-}$ Jeg3 cells expressing flag-tagged WT-ARHGAP18-flag (rescue) or the control flag vector. XY images above; side projections showing apical/basolateral localization below. Scale bars 10 µm. (**E**) Immunofluorescence confocal images of ARHGAP18$^{-/-}$ Jeg3 cells expressing ARHGAP18(319–523) GAP domain only, a point mutant GAP-dead variant or a chimera variant of the GAP domain of ARHGAP18(319–523) linked to the ezrin binding motif of EBP50(486–523). Expressing the GAP domain alone does not target to microvilli, disrupts microvilli formation and ezrin targeting to the apical surface. The GAP-dead variant is localized to microvilli and apical ezrin but does not rescue the knockout microvilli phenotype (zoomed inset). GAP-flag-EBP50(tail) targets to microvilli (inset), is sufficient to reduce microvilli abundance in the ARHGAP18$^{-/-}$ cells, and ensures ezrin localization to the apical surface. Scale bars 10 µm, insets 2 µm. (**F**) Quantification of the density of microvilli on the surface of ARHGAP18$^{-/-}$ cells expressing the conditions shown in (**C–E**). Microvilli counts per cell were normalized to the area of each cell to account for cell size variability. ARHGAP18$^{-/-}$+ARHGAP18-flag condition duplicated from *Figure 2F* for comparison between conditions. Bars represents mean ± SEM; n ≥ 3; *t*-test; *p≤0.05. Side projections expanded in Z-dimension for clarity.

**Video 1.** Time-lapse movie of live WT Jeg3 vs. ARHGAP18⁻ᐟ⁻ cells transiently expressing GFP-EBP50 which targets to microvilli. Each frame taken at 1 min intervals. ARHGAP18⁻ᐟ⁻ cells have more but smaller microvilli, which turn over more rapidly than in the WT condition. Scale bars 5 μm.

https://elifesciences.org/articles/83526/figures#video1

**Video 2.** Time-lapse movie of live WT Jeg3 vs. ARHGAP18⁻ᐟ⁻ cells transiently expressing GFP-EBP50. Each frame taken at 1 min intervals. Tracking of Individual microvilli is shown. Scale bars 5 μm.

https://elifesciences.org/articles/83526/figures#video2

domain of Ezrin (1–296) and identified a predicted eight amino acid region within ARHGAP18 from V10 to S17 likely to interact with ezrin (*Figure 4A*). Deletion of this region (ARHGAP18-(Δ10–17)-Flag) resulted in a total loss of microvilli localization by ARHGAP18 (*Figure 4B*). In agreement with our pulldown results (*Figure 1B and C*), the predicted binding region of A14-I20 on the ezrin-FERM would be sterically blocked by the intramolecular binding of the ezrin-CTD to the ezrin-FERM in the inactivated form (*Figure 1A*; *Pearson et al., 2000*). These results support the conclusion that ARHGAP18 is targeted to areas of active ezrin via a ezrin-FERM binding motif within ARHGAP18 residues V10 to S17 (*Figure 4A and B*).

## Ezrin binding to ARHGAP18 enhances its GAP activity to locally reduce active RhoA inside microvilli

We sought to understand how ezrin binding may influence ARHGAP18's biochemical activity toward RhoA's GTPase activity. Using an in vitro RhoA GAP colorimetric assay that measures free phosphate (Pi) released by RhoA-dependent hydrolysis of GTP to GDP, the intrinsic GAP activity of recombinant RhoA was found to be enhanced about fourfold in the presence of purified ARHGAP18 (*Figure 4C and D*). This activity was about half the rate obtained with the purified control Rho GTPase-activating protein 1(p50RhoA GAP) (*Figure 4D*). When the ezrin FERM domain was included with ARHGAP18, the GAP activity was enhanced about twofold to the level of the control p50RhoA GAP. We conclude that the binding of ARHGAP18 to ezrin's FERM domain enhances its catalytic activity, thereby increasing its activity in microvilli where active ezrin exists. Activation in this way would limit the potency of its GAP activity to areas of the cell where active, open ezrin exists.

Loss of ARHGAP18 should increase the levels of active RhoA at the apical surface of the cells, but due to the presence of numerous potentially redundant GAPs, it was not clear what effect this might have. Western blotting on cell lysates was used to test the expression and activation of RhoA downstream targets. Expression of total cellular levels of RhoA itself remained unchanged in the knockout cells (*Figure 4E and F*). One result of RhoA activation is to increase the levels of phosphorylated myosin regulatory light chain (pMLC). We measured an increase in pMLC in ARHGAP18⁻ᐟ⁻ cells (*Figure 4E and F*). While whole-cell RhoA levels did not change within detectable levels via western blotting, we predicted that local levels of active RhoA would increase at the apical surface in the absence of ARHGAP18. To visualize the active fraction of RhoA locally at the apical surface, we expressed a biosensor, AHPH-GFP, that preferentially binds active RhoA-GTP (*Piekny and Glotzer, 2008*). Utilizing confocal microscopy to image the apical section of Jeg3 cells, we observed that in wildtype cells the GFP signal was localized to cell–cell junctions as reported previously (*Priya et al., 2015*; *Figure 4G*). However, expression of the biosensor in ARHGAP18⁻ᐟ⁻ cells showed a reorganization of RhoA-GTP to the apical surface of the cell, though some signal remained at junctions (*Figure 4G*). We quantified these results over multiple cells within

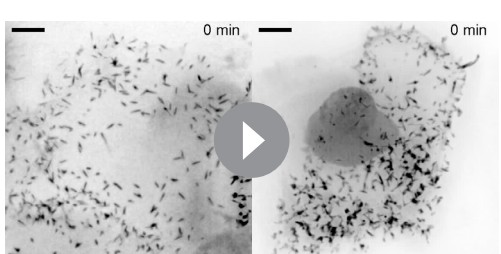

**Video 3.** Time-lapse movie of live WT Jeg3 vs. ARHGAP18⁻ᐟ⁻ cells transiently expressing GFP-EBP50. Each frame taken at 1 min intervals. Still frames of selected individual microvilli shown in *Figure 3A*. Scale bars 5 μm.

https://elifesciences.org/articles/83526/figures#video3

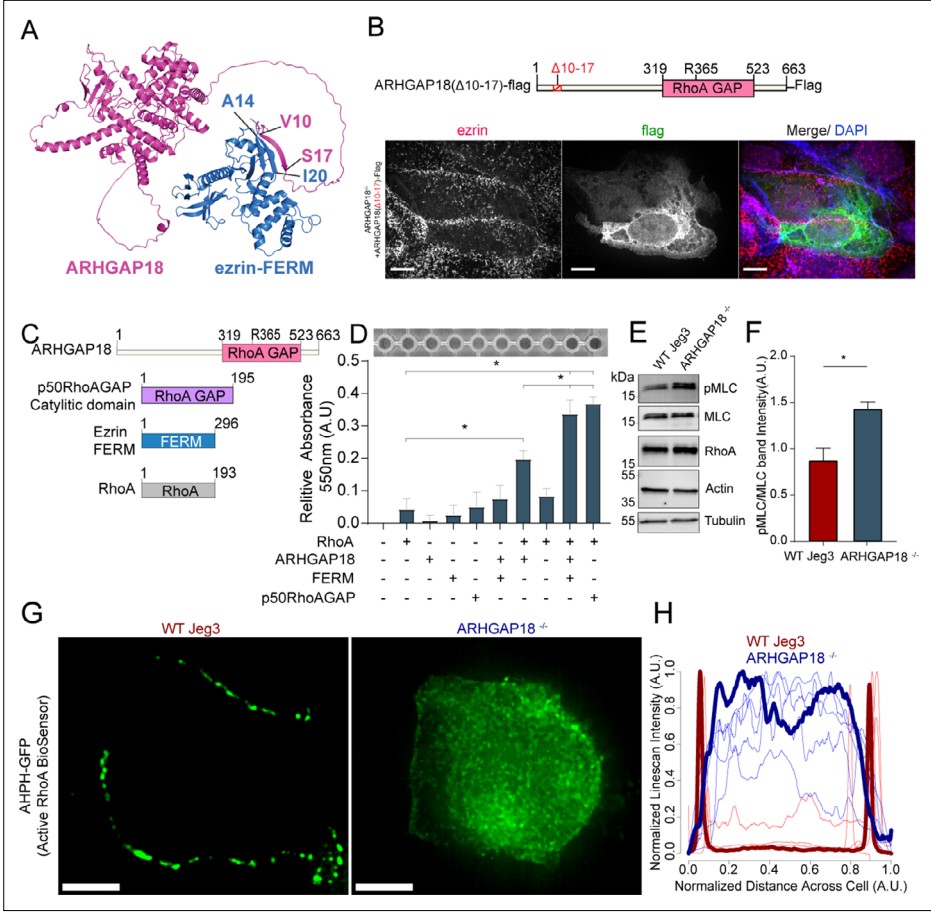

**Figure 4.** Ezrin binding to ARHGAP18 enhances GAP activity excluding active RhoA inside microvilli.
(**A**) AlphaFold2 structural interaction prediction of full-length human ARHGAP18 (magenta) with human ezrin-FERM domain (blue). (**B**) Schematic of and of ARHGAP18(Δ10–17)-flag construct and fluorescent images of ARHGAP18$^{-/-}$ Jeg3 cells expressing the construct which localizes diffusely throughout the cytoplasm. (**C**) Cartoon schematic of purified proteins used in (**D**). (**D**) Representative image of RhoA GAP assay and graph of combined results from multiple assays. Each assay normalized to the buffer only condition. Bars represents mean ± SEM; n = 4; *t*-test; *p≤0.05. (**E**) Western blotting indicating changes in phosphorylated myosin regulatory light chain (pMLC) in response to altered expression of ARHGAP18 in Jeg3 cells. (**F**) Quantification of pMLC over total MLC from replicates of the data in (**E**). (**F**) Bars represents mean ± SEM; n = 4; *t*-test; *p≤0.05. (**G**) Z-projection confocal fluorescent images of active RhoA biosensor. Scale bars 10 μm. (**H**) Line scan intensity profiles of RhoA biosensor across single WT (red) and ARHGAP18$^{-/-}$ (blue) cells. Line scans from the cells shown in (**G**) are bold while replicate cells shown as thinner lines (n = 11 cells).

The online version of this article includes the following source data for figure 4:

**Source data 1.** Full western blot images from all blots shown in *Figure 4*.

each condition using an intensity line scan across the entire apical surface which showed that active RhoA biosensor signal dramatically reorganizes across the apical surface within ARHGAP18$^{-/-}$ cells (*Figure 4H*).

We aimed to further understand the effect of ARHGAP18$^{-/-}$ cells' aberrant active RhoA and increased pMLC activation at the apical surface (*Figure 4F and G*). Phosphorylation of the myosin light chain leads to activation and production of force or tension from non-muscle myosin motors on actin filaments. We previously characterized a cellular force indentation technique using an atomic force microscopy probe tip to directly measure stiffness changes at the apical surface in response to alterations in local RhoA signaling at microvilli and the apical surface (*Zaman et al., 2021*). Using the force indentation technique, we confirmed that the Young's moduli of the apical surface of cells lacking ARHGAP18 were approximately twice the stiffness (2.0 ± 0.1 kPa) on average compared to wildtype cells (1.1 ± 0.1 kPa) (*Figure 5A and B*). These data indicate that the reorganization of active

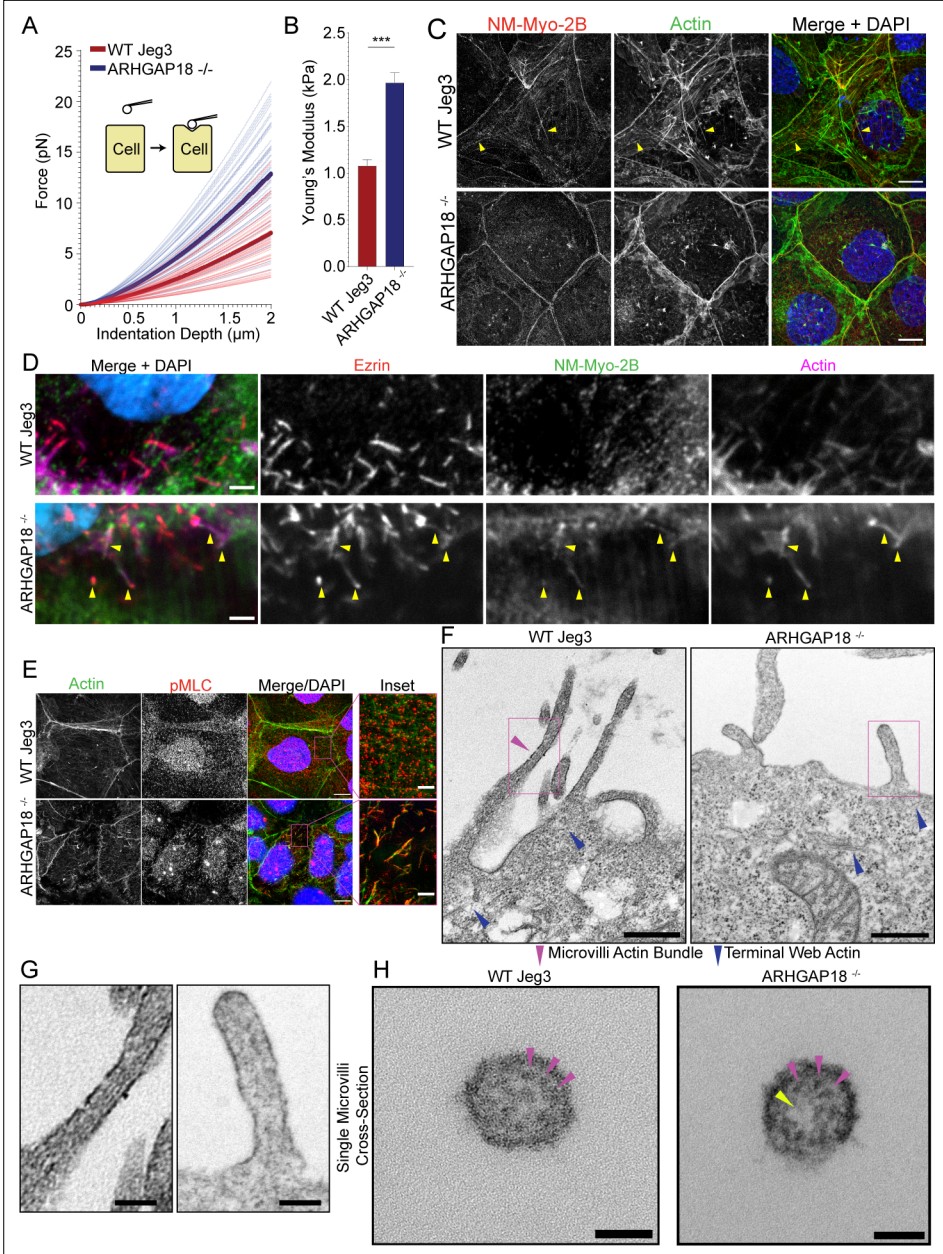

**Figure 5.** ARHGAP18 regulates actomyosin activation and organization in microvilli. (**A**) Graph of force as a function of indentation depth. As the probe tip is indented against the cell surface, the force increases following a Hertzian model. Each trace indicates the fit curve from an individual cell indentation profile comparing ARHGAP18-/- cells (blue) to WT Jeg3 cells (red). The bold lines represent the average curve for each condition. (**B**) Average Young's modulus WT = 1.1 ± 0.1 kPa; ARHGAP18-/- = 2.0 ± 0.1 kPa (mean ± SEM); error bars represent the SEM. Significance: *t*-test; p<0.0001; n = 49 WT cells and 54 ARHGAP18-/- cells. (**C**) Representative maximum Z-projections of immunofluorescence structured illumination microscopy (SIM) images of fixes cells showing non-muscle myosin-2B localization (yellow arrows show contractile fibers). Scale bars 10 μm. (**D**) Immunofluorescence staining of ezrin and non-muscle myosin-2B showing colocalization of actin, ezrin, and non-muscle myosin to microvilli in cells lacking ARHGAP18 imaged using SoRa confocal Z-projected slices of the apical surface. Scale bars 2 μm. (**E**) Immunofluorescence staining of phosphorylated myosin regulatory light chain (pMLC) in WT Jeg3 cells vs ARHGAP18-/- cells inset showing localization to the microvilli in the KO cells. Scale bars 10 μm; inset 2 μm. (**F**) Representative transmission electron microscopy (TEM) of actin at the microvilli core bundle (magenta arrowhead) to terminal web (blue arrowhead) interface. The actin core bundle in WT cells is made from aligned tightly bundled actin filaments; however, the actin in microvilli of cells lacking ARHGAP18 is less organized and aligned bundled actin is not visible. The core bundle embeds into the terminal web actin (blue arrowheads) which

*Figure 5 continued on next page*

*Figure 5 continued*

is made from a delicate network of actin in both conditions and has no visibly apparent differences between conditions. Scale bars 500 nm. (**G**) Zoom-in of the boxed areas from (**F**). Scale bars 125 nm. (**H**) Cross section of a single microvillus where individual actin filaments making up the core actin bundle are resolved (magenta arrowheads). The ARHGAP18$^{-/-}$ microvillus shows a central region of lighter electron density or 'hole' where there are no actin filaments (yellow arrowhead). In contrast, the WT cell microvillus shows actin filaments throughout the internal volume. Scale bars 50 nm.

The online version of this article includes the following figure supplement(s) for figure 5:

**Figure supplement 1.** Comparison of phospho-myosin light chain, non-muscle myosin-2A, B, or C, and actin in WT Jeg3 or ARHGAP18$^{-/-}$ cells.

**Figure supplement 2.** Comparison of non-muscle myosin-2B, and actin in WT Jeg3, LOK$^{-/-}$SLK$^{-/-}$ ARHGAP18$^{-/-}$, and ARHGAP18 rescue with empty vector control.

RhoA visualized at the apical surface of ARHGAP18$^{-/-}$ cells was altering the mechanical properties of the local actomyosin cytoskeletal networks with microvilli or just below the apical membrane. Immunofluorescence detection of pMLC is typically observed in contractile actomyosin bundles or networks where non-muscle myosin-2 is active. We performed immunofluorescent confocal imaging of pMLC colocalized with each of the three non-muscle myosins-2 isoforms A, B, or C (*Figure 5—figure supplement 1*). The side-by-side comparative localization of these non-muscle myosin-2 isoforms within WT Jeg-3 cells alone represents, to our knowledge, a first for human placental cells. In agreement with previously presented data by our group and others (*Chinowsky et al., 2020*; *Zaman et al., 2021*), isoforms 2A and 2B appear to localize to nearly identical structures including large contractile bundles and cell–cell junctions. Non-muscle myosin-2C does not readily localize with stress fibers or larger contractile bundles in agreement with recent characterizations (*Chinowsky et al., 2020*). We then compared the localizations of non-muscle myosin-2 between WT and ARHGAP18-deficient cells (*Figure 5C*, *Figure 5—figure supplement 1*).

When we imagined the apical surface of wildtype Jeg3 cells for non-muscle myosin-2B, we observed few long contractile bundles in wildtype cells (*Figure 5C*). In the ARHGAP18$^{-/-}$ cells, we observed the non-muscle myosin-2B was reorganized into smaller punctate and microvillar-like structures and a total absence of any larger contractile bundles (*Figure 5C and D*) in agreement with ARHGAP18 knockdown results from medaka fish (*Porazinski et al., 2015*). This phenotype could be rescued by the expression of ARHGAP18-flag in the ARHGAP18$^{-/-}$ cells (*Figure 5—figure supplement 2*). We then used an antibody against pMLC to observe the activated fraction of non-muscle myosin-2 at the apical surface. In wildtype cells, pMLC was found along the cell–cell junctions and in punctate structures near the terminal web actin, just below the apical membrane (*Figure 5E*, inset). This observation agrees with reports of active non-muscle myosin-2 localizing to the region directly below the microvilli within the terminal web of actin, where it depolymerizes the core bundle and regulates microvilli length (*Meenderink et al., 2019*). However, pMLC in the ARHGAP18$^{-/-}$ cells was inappropriately localized inside microvilli where it overlapped with the actin core bundles (*Figure 5E*, inset). Thus, ARHGAP18 is necessary to modulate local RhoA activity in microvilli to ensure that myosins are not inappropriately assembled there, while ensuring sufficient RhoA levels to locally activate LOK and SLK to phosphorylate ezrin.

We hypothesized that the increased non-muscle myosin-2 activity within the microvilli of ARHGAP18-deficient cells was disrupting the organization of actin within the microvilli. The individual actin filaments of the microvilli bundle and terminal web are approximately 7 nm in width and not resolvable using light microscopy techniques. Thus, we employed transmission electron microscopy (TEM) on embedded sections of the apical surface of Jeg3 cells where the microvilli core bundle interacts directly with the terminal web actin (*Figure 5F*). In the wildtype cells, we observed a bundled actin within the microvilli (*Figure 5F*, magenta arrowhead, and G). In cells lacking ARHGAP18, we observed the dramatically shortened microvilli as characterized above (*Figure 2G*). Actin filaments within the microvilli of ARHGAP18-deficient cells were observed; however, they lacked the distinct parallel aligned actin core bundles found in the WT cells. The terminal web of actin in the cultured cells was difficult to resolve in our TEM images, which is not surprising given that fluorescent imaging of these actin networks indicates it is comprised of a comparatively sparse network of individual actin filaments (*Figure 5F*). However, individual and interwoven networks of actin beneath the apical surface

and microvilli were observed (*Figure 5F*, blue arrowheads). To investigate the alterations to the actin bundle organization within individual microvilli further, we imaged cross sections of single microvilli at a magnification of 60,000× using TEM. Under these conditions, the individual actin filaments within a single microvillus were resolved (*Figure 5H*). Within the microvilli of cells lacking ARHGAP18, the actin filament core bundles were less dense than in wildtype cells (*Figure 5G and H*) with some knockout cell microvilli showing electron density holes where a region of the microvillus was devoid of filaments (*Figure 5H*, yellow arrowhead). These results suggest that the increased non-muscle myosin-2 activity in ARHGAP18-deficient cells results in inappropriate disassembly of actin filaments within microvilli.

These results collectively promote a model where ARHGAP18 acts as a negative regulator of RhoA locally within regions where ERMs are active (*Figure 6*). This activity regulates non-muscle myosin-2 activation and the actin at the apical surface and within microvilli. ARHGAP18 is regulated downstream of LOK/SLK, where activation of ERMs both localizes ARHGAP18 and enhances its GAP activity.

## Discussion

### ARHGAP18 limits the activity of RhoA in microvilli through binding to active ezrin

This study reveals that active ezrin directly recruits and activates ARHGAP18 to microvilli to locally regulate RhoA activity for appropriate ezrin phosphorylation and limit apical actomyosin organization. In earlier work, we characterized the Jeg3 cells used here lacking expression of ERMs or LOK/SLK and demonstrated that LOK/SLK are the major kinases that phosphorylate ERM proteins in humans (*Zaman et al., 2021*). We also showed that a constitutively active form of RhoA directly binds LOK. An independent proteomic screen identifying Rho GTPase family protein interactors showed that RhoA directly binds SLK, leading to dimerization and autophosphorylation (*Bagci et al., 2020*). These results define an activation pathway where RhoA activates LOK/SLK that then activates ERMs. Additionally, we found that in the absence of LOK/SLK or ERMs, the level of active RhoA was elevated, indicating that active ERMs in some way downregulate active RhoA (*Zaman et al., 2021*). These findings indicated a missing negative regulatory feedback loop through an unknown effector, an idea supported by studies in the genetically simpler fly model (*Speck et al., 2003*). As introduced earlier, ARHGAP18 was identified both in our proteomic screen for active ezrin interactors (*Viswanatha et al., 2013*) and its probable fly homolog Conundrum in a two-hybrid screen with fly moesin (*Neisch et al., 2013*). We find that ARHGAP18 binding to active ezrin activates the RhoA-GAP activity of ARHGAP18, and this activity is necessary for appropriate regulation of the apical domain (*Figure 6*).

### Potential mechanism of apical regulation through RhoA, LOK/SLK, ERMs, and ARHGAP18

Cells lacking either LOK/SLK or all ERMs have no apical microvilli, underscoring the important roles of these two classes of proteins (*Zaman et al., 2021*). ERMs interact through their FERM domain with $PI(4,5)P_2$, which is preferentially enriched at the plasma membrane. Biochemical reconstitution studies show that activation of ERMs by LOK or SLK then occurs through a coincidence detection mechanism whereby ERMs are activated through phosphorylation only when bound simultaneously to LOK/SLK and $PI(4,5)P_2$ (*Pelaseyed et al., 2017*). Once active, the ERMs serve a required function of linking the plasma membrane to the actin core bundle formed within the microvilli. Our data show that ARHGAP18 is recruited to the microvilli through its preferential binding to the open active ERM's FERM domain (*Figure 1C*). The localization of ARHGAP18 in microvilli is particularly interesting as we are unaware of any other GAP or GEF that localizes specifically to microvilli. An independent screen of brush border enterocyte proteins also did not identify one; however, the study did identify Rho precursors to isoforms A, B, C, and G (*McConnell et al., 2011*). These results suggest RhoA is not excluded from microvilli but instead kept modulated locally by ARHGAP18. RhoA is active just below the microvilli within the terminal web of actin (*Shaw et al., 1998*), and ARHGAP18's ability to enhance its GAP activity when bound exclusively to active, open ERMs within microvilli indicates that the microvilli are regulated as a separate RhoA microdomain on the individual microvilli scale (200–1000 nm). This model may represent a broader mechanism employed to regulate the spatio-temporal coordination of Rho GTPases across multiple cell surface protrusions through various GEFs and GAPs. For example, the ARHGAP35 gene product p190A has been reported to localize to cilia

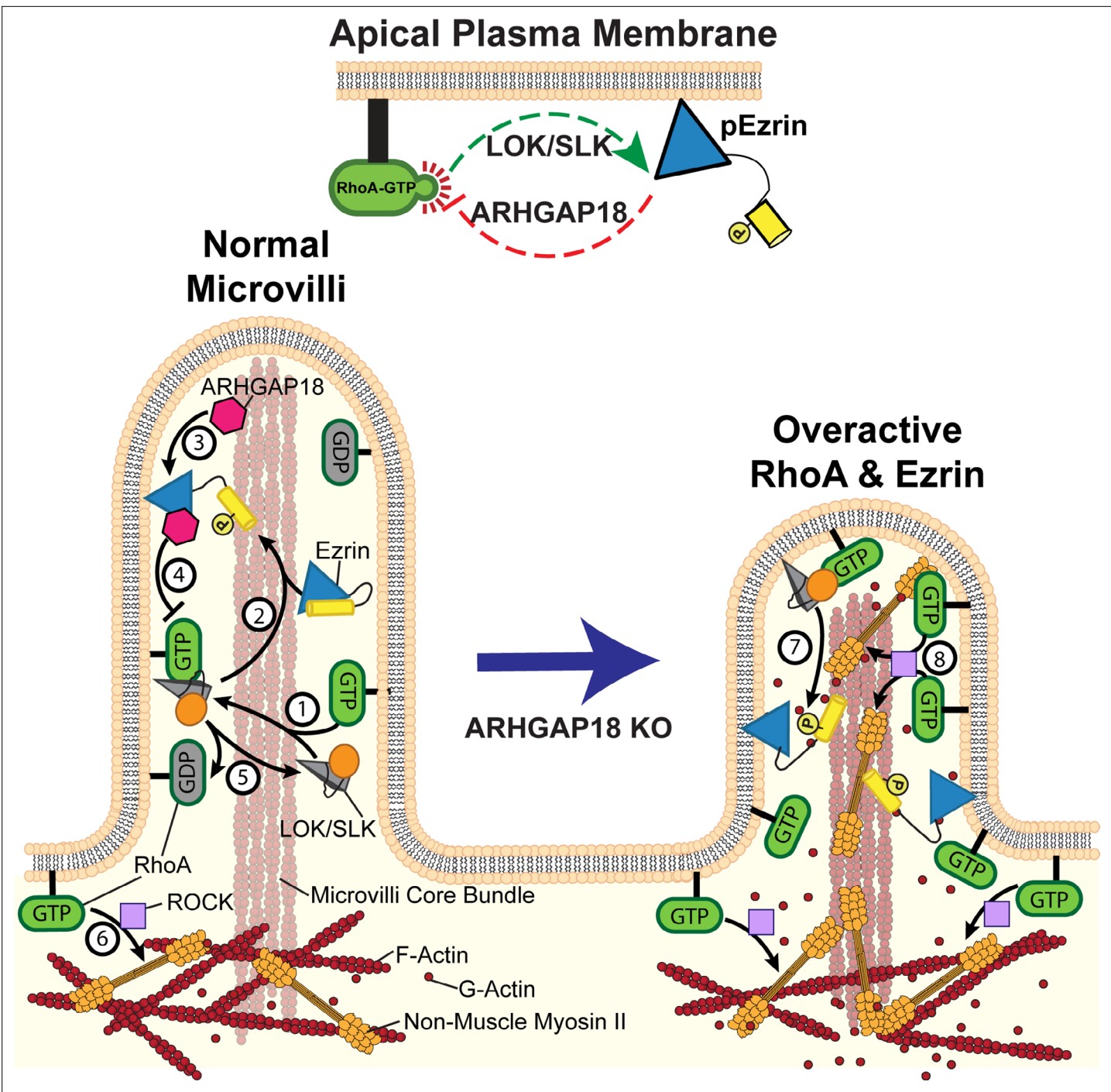

**Figure 6.** Model of ezrin and RhoA feedback loops in microvilli. (1) Active RhoA binds and activates LOK and SLK to activate ezrin through their kinase domain (orange circle). These kinases are specifically localized to the microvilli of epithelial cells. (2) In cooperation with $PIP_2$, the kinases phosphorylate and activate ezrin. Once active, ezrin links the plasma membrane to the actin core bundle within microvilli and also makes available a binding site for ARHGAP18. (3) ARHGAP18 binds the FERM domain of ezrin and this activates its RhoA-GAP activity. (4) This activity locally reduces the concentration of active-RhoA in microvilli, thereby modulating the ability of RhoA to activate LOK to phosphorylate ezrin. (5) Inactive Rho-GDP cannot bind or activate LOK kinase, thus completing a negative feedback system (top center of figure). (6) In the terminal web, RhoA-GTP activates the assembly of non-muscle myosin-2 through the ROCK pathway. In ARHGAP18 knockout cells, excessive active RhoA in the microvilli (7) overactivates ezrin through the action of LOK/SLK and (8) promotes inappropriate assembly of myosin-2 through the activity of ROCK. These changes result in increased microvilli formation (through ERMs) and disruption of the F-actin core bundle within microvilli (through non-muscle myosin-2).

where Rho activity is required for proper cilia formation (*Stewart et al., 2016*), RhoA acts specifically at sites of protrusion initiation in lamellipodia (*Machacek et al., 2009*) and SRGAP1 controls podocyte morphology through localized regulation of Rho activity (*Rogg et al., 2021*).

Cells lacking ARHGAP18 exhibit two simultaneous alterations to microvilli regulation. First, cells deficient in ARHGAP18 have enhanced RhoA activity at the apical surface that increases ezrin phosphorylation. This enhanced level of active ezrin increases the number of microvilli, showing that ARHGAP18 negatively regulates ezrin activation through the RhoA/LOK/SLK pathway. Second, RhoA also regulates actin and non-muscle myosin-2 through the ROCK pathway. The loss of ARHGAP18 and subsequent increase in RhoA activation at the apical surface result in increased activation of non-muscle myosin-2 (*Figure 4E–H*; *Maeda et al., 2011*; *Porazinski et al., 2015*; *Zaman et al., 2021*). In WT cells, it has been proposed that increased activation of non-muscle myosin-2 results in depolymerization of the microvilli core bundle within the terminal web of actin (*Meenderink et al., 2019*). For this model to be correct, non-muscle myosin-2 activity must be excluded from within the microvilli itself, a conclusion supported by this report. In cells lacking ARHGAP18, non-muscle myosin-2 activity is inappropriately expanded to occur inside the microvilli through RhoA activation there. This aberrant myosin activation results in the increased turnover of microvilli (*Figure 3A*) as the actin core bundles are depolymerized more quickly by increased non-muscle myosin-2 activity. Thus, by combining these two effects in ARHGAP18[-/-] cells, the number of microvilli is enhanced as a result of increased active ERMs, yet microvilli length is decreased as a result of increased non-muscle myosin-2 activity. Our live-cell tracking of GFP-EBP50-labeled microvilli, SEM, and TEM imaging data all indicates that it is possible for the microvilli in cells lacking ARHGAP18 to achieve full length. However, the balance between turnover and maintenance is altered, reducing the likelihood of achieving full length. It is possible that the increased turnover of microvilli core bundle actin results in the freeing of actin monomers that would normally be sequestered into the longer microvilli actin found in WT cells (*Figure 6*). The incorporation of these actin monomers into nearby structures may explain the increased apical stiffness measured in ARHGAP18-deficient cells (*Figure 5A and B*). Recent data suggests that microvilli biogenesis involves at a minimum the combined efforts of EPS8, IRTKS, and ERMs working in concert and future work will be required to define how ARHGAP18 may interact with the activities of EPS8 or IRTKS (*Gaeta et al., 2021*).

It is of particular interest to note that the broad mechanism proposed here was correctly speculated previously. To explain the phenotypes observed in *Drosophila* moesin mutants, Neisch et al. suggested that it is the joint effects of increased RhoA-dependent apical actomyosin contractility and loss of moesin-mediated membrane–cytoskeleton cross-linking activity (*Neisch et al., 2013*). Our data support this model within the human system and additionally provide characterization and localization of the specific interactors facilitating these signaling interactions at the apical surface.

## Implications for the local regulation of Rho proteins

A major gap in our understanding of active Rho GTPases, which are free to diffuse in the plasma membrane, is how they select and discriminate between their many effectors (reviewed in *Denk-Lobnig and Martin, 2019*). This discrimination could be mediated by effectors with different affinities for active Rho, or their abundance, or location, or any combination of these. In the simple case investigated here, where active RhoA must discriminate between the LOK/SLK pathway to activate ERM proteins and the ROCK pathway that leads to myosin activation, we now have some insights into how this may occur. In the case of LOK/SLK, these effector kinases are specifically targeted to the apical membrane by the C-terminal half of the molecule (*Pelaseyed et al., 2017*; *Viswanatha et al., 2012*). However, as revealed by this study, this selective targeting of the kinase is not sufficient for functional discrimination by active RhoA as loss of ARHGAP18 results in the overactivation of ezrin and inappropriate assembly of non-muscle myosin-2. Thus, targeting of ARHGAP18 to microvilli by ezrin is necessary to locally tune the level of active RhoA to ensure that sufficient ezrin is phosphorylated while ensuring non-muscle myosin-2 is not activated. An attractive mechanism could be to locally reduce the level of active RhoA to a level commensurate with the local microvillar abundance of LOK/SLK, thereby driving selectivity by mass action. As part of this regulation, the level of active ezrin is itself determined by the activity of RhoA through LOK/SLK and that, in turn, determines the recruitment and activity of ARHGAP18, which then negatively regulates RhoA (*Figure 6*). This auto-regulatory loop would ensure that excessive RhoA activity is immediately modulated down to the

appropriate level, and too little activity would be modulated up to the appropriate level. This highly localized auto-regulatory loop thereby fine-tunes RhoA activity for the appropriate morphogenesis of the apical membrane. We expect that similar fine-tuning of the activity of other small GTPases by downstream targets will emerge as more is learned about the GAP proteins that vastly outnumber the small number of GTPases they regulate.

# Materials and methods

**Key resources table**

| Reagent type (species) or resource | Designation | Source or reference | Identifiers | Additional information |
|---|---|---|---|---|
| Gene (*Homo sapiens*) | ARHGAP18 | ensembl.org | ENSG00000146376 | |
| Strain, strain background (*H. sapiens*) | OneShot TOP10 (DH10B) | Thermo Fisher | C404010 | Chemically competent cells |
| Cell line (*H. sapiens*) | Jeg-3 | ATCC.org | Htb-36 | |
| Cell line (*H. sapiens*) | DLD-1 | ATCC.org | CCL-221 | |
| Cell line (*H. sapiens*) | Jeg-3 LOK$^{-/-}$SLK$^{-/-}$ | *Zaman et al., 2021* | LOK$^{-/-}$SLK$^{-/-}$ | LOK and SLK knock out cells |
| Cell line (*H. sapiens*) | Jeg-3 ARHGAP18$^{-/-}$ | This paper | ARHGAP18$^{-/-}$ Clone 2–4 | ARHGAP18 knock out cells |
| Antibody | Anti-ARHGAP18 rabbit (polyclonal) | This paper | B116 | Antigen full-length human ARHGAP18 (1:1000 for WB) |
| Antibody | Anti-ezrin rabbit (polyclonal) | *Bretscher, 1989* | B64 | Antigen full-length human Ezrin (1:1000 for WB) |
| Antibody | Anti-phospho-ERM antibody rabbit (polyclonal) | *Hanono et al., 2006* | B81 | Detects pT567 on ezrin, radixin, and moesin (1:1000 for WB) |
| Antibody | Anti-phospho-myosin light chain, rabbit (polyclonal) | Cell Signaling Technology | 3674 | Against phospho-Thr18/Ser19 (1:500 for WB) |
| Antibody | Anti-non-muscle myosin-2B, rabbit (polyclonal) | BioLegend | 909902 | (1:100 for WB) |
| Recombinant DNA reagent | Human ARHGAP18 | Harvard Plasma Database | HsCD00379004 | cDNA of ARHGAP18 |
| Recombinant DNA reagent | EBP50-GFP | *Garbett and Bretscher, 2012* | ATL 1469 | Backbone pEGFP-C2 |
| Recombinant DNA reagent | AHPH-GFP | *Piekny and Glotzer, 2008* | Addgene; 68026 | Backbone pEGFP-C1 |
| Recombinant DNA reagent | SUMO-ARHGAP18 | This paper | ATL 4748 | Backbone: pE-SUMO |
| Recombinant DNA reagent | ARHGAP18-flag | This paper | ATL 3160 | Backbone: pQCXIP |
| Recombinant DNA reagent | ARHGAP18-pLentiCrispr-V2 | This paper | ATL 4740 | Backbone: lentiCrispr-V2 |
| Commercial assay or kit | RhoA GAP assay kit | Cytoskeleton Inc | BK105 | |

## Western blotting

Western blotting was done using 12–8% split SDS-PAGE gels except for proteins less than 30 kDa that used 14% SDS-PAGE gels for better resolution. The gels were transferred at 16 V to 0.45 µm pore size polyvinylidene difluoride (PVDF) membranes (EMD Millipore Immobilon-FL; Cat# IPFL85R) or 0.2 µm pore size PVDF membranes (EMD Millipore Immobilon-P$^{SQ}$; Cat# ISEQ08130) for smaller proteins less than 30 kDa. Cell lysates were washed and harvested with warm Laemmli sample buffer (70°C) by scraping and then boiling. Membranes were blocked with 5% nonfat milk in Tris-buffered saline with 0.1% tween (TBS-T), except for phosphorylation-specific antibodies which used Immobilon Block – PO phosphoprotein blocking buffer (EMD Millipore; Cat# WBAVDP001). Primary antibodies diluted in either 5% BSA in TBS-T or Immobilon Block – PO were incubated with the membrane at room temperature (RT) for 1 hr or at 4°C overnight. The signal was detected using fluorescent secondary antibodies, anti-mouse Alexa Fluor IRDye 680RD, and anti-rabbit Alexa Fluor IRDye 800CW (LI-COR Biosciences; Cat# 926-68070 and 926-32211) at 1:10,000 in 5% nonfat milk.

Blots were imaged using a Bio-Rad ChemiDoc, and band intensities were determined with ImageJ's gel analysis package by normalizing band intensity against either cell housekeeping proteins (e.g., tubulin) or the unphosphorylated total protein of interest detected in the input sample, then calculating relative intensities. Gel band intensities were then input to GraphPad Prism for quantification and statistical analysis.

## ARHGAP18 and ezrin iFlag pulldown assays

To determine the interaction between ARHGAP18 and ezrin, pulldown assays were performed using cell lysate from Jeg3 wildtype and Jeg3 LOK$^{-/-}$SLK$^{-/-}$ cells. Transfections were performed using PEI MAX polyethylenimine reagent (Polysciences; Cat# 24765). After transfection, cells were washed with phosphate-buffered saline (PBS) and harvested with lysis buffer (25 mM Tris, 5% glycerol, 150 mM NaCl, 50 mM NaF, 0.1 mM Na$_3$VO$_4$, 10mM βGP, 0.2% Triton X-100, 250 mM calyculin A, 1 mM DTT, 1× cOmplete Protease Inhibitor Cocktail [Roche; Cat# 11836153001]) by scraping. Lysates were then sonicated, and any insoluble material was centrifuged at 8000 × $g$ for 10 min at 4°C. Before incubating with the cell lysates, SUMO-ARHGAP18 NiNTA or M2-Flag resin beads were equilibrated and washed into lysis buffer. The sample of the supernatant was taken for input, then the rest was added to the SUMO-ARHGAP18 NiNTA or M2-Flag beads and nutated for 3 hr at 4°C. After incubation, the beads were pelleted and washed four times before boiling in 40 μL 2× Laemmli sample buffer.

## Cell culture

Jeg3 cells originally obtained from ATCC.org (Cat# Htb-36) were maintained in a humidified incubator at 37°C and 5% CO$_2$. Jeg3 cells were maintained in 1× MEM (Thermo Fisher Scientific; Cat# 10370088) with penicillin/streptomycin (Thermo Fisher Scientific; Cat# 15070063), 10% fetal bovine serum (FBS; Thermo Fisher Scientific; Cat# 26140079), and GlutaMAX (Thermo Fisher Scientific; Cat# 35050061). Cultures were maintained on Corning 100 × 20 mm Petri-style TC-treated culture dishes (Cat# 430167). Transient transfections were accomplished using a PEI MAX polyethylenimine reagent (Polysciences; Cat# 24765). DLD-1 cells originally obtained from ATCC.org (Cat# CCL-221) were handled identically with the exception of using DMEM media. All cell lines were checked for mycoplasma monthly by DAPI staining.

## CRISPR knockout lines

CRISPR analysis tools from Benchling Inc (San Francisco, CA) were used to design single-guide RNAs (sgRNAs). Guides were cloned into puromycin-resistant pLenti-CRISPRV2 (Addgene; Cat# 49535) as described in *Sanjana et al., 2014*. We created two potential guides against human ARHGAP18. We also created two distinct KO lineages using separate CRISPR/CAS9 target guides both near the start of the coding region in exon at 5'-CTAACAGCCTACCACCCCAG-3' and 5'-CAGCGGCAAGGACCAG ACCG-3'. We first identified the core phonotypes of the ARHGAP18KO cells, numerous shortened microvilli, in screening experiments in both populations of KOs. Both sgRNA sequences used against human ARHGAP18 were transfected into 293TN cells with psPAX3 and pCMV-VSV-G (a gift from Jan Lammerding, Weill Institute for Cell and Molecular Biology, Cornell University, Ithaca, NY) for 48–72 hr before virus collection. Jeg3 cells were then transduced with viral media supplemented with Polybrene to 8 μg/mL twice a day for 2 d prior to puromycin selection at 2 μg/mL. Single-cell sorting was performed and then expanded in puromycin selection before screening by western blotting. The 5'-CAGCGGCAAGGACCAGACCG-3' guide sequence produced more uniform phenotype expression and was used for all experiments.

## Microvilli quantification and tracking

To determine whether ARHGAP18's presence influences the abundance of microvilli in Jeg3 cells, cells were transfected with wildtype ezrin as a marker for microvillar abundance. The abundance of microvilli was quantified using ImageJ's cell counter toolkit, and total microvilli counts for each cell were normalized to their respective cell areas to account for variability in cell sizes. Values were then input to GraphPad Prism for quantification and statistical analysis. Tracking of microvilli lifetimes was done using the ImageJ plugin MTrackj originally developed in *Meijering et al., 2012*.

## Fixed cell preparation

Jeg3 cells were grown on glass coverslips prior to fixing cells for immunofluorescence. Cells were washed with PBS before being fixed in 3.7% formaldehyde in PBS for 10 min at RT, then washed in PBS three times. Cell membranes were permeated using 0.2% Triton X-100 diluted in PBS. Blocking was performed using 2% FBS in PBS for 10 min. Primary and secondary antibodies were incubated on the cells for 1 hr at RT each and were diluted in the 2% FBS blocking solution. To visualize actin or the nucleus, Alexa Fluor 647 Phalloidin at 1:25 dilution in methanol and DAPI at 1:10,000 dilution in $H_2O$ were included with the secondary antibody incubation. The cells were mounted with either SlowFade Diamond Antifade (Thermo Fisher Scientific; Cat# S36967) or ProLong Diamond Antifade (Thermo Fisher Scientific; P36965). For AHPH-GFP (Addgene; Cat# 68026), active RhoA biosensor cells were fixed using 10% trichloroacetic acid (TCA) instead of formaldehyde. AHPH-GFP quantification data was measured using the ImageJ built-in intensity profile feature. Intensity vs. distance plots were normalized and then passed through a nine-point sliding-average window to reduce signal noise in the traces. Individual intensity plots were then overlaid using R-Programming language (*R Development Core Team, 2022*) and RStudio integrated development environment (*RStudio Team, 2022*).

## Scanning electron microscopy (SEM) and transmission electron microscopy (TEM)

SEM and TEM imaging was performed at the Microscopy Imaging Center at the University of Vermont (Burlington, VT, RRID#:SCR_018821). Cells were grown on Thermanox coverslips fixed using Karnovski's fixative (2.5% glutaraldehyde, 2% paraformaldehyde 0.1 M cacodylate buffer, pH 7.2) for 60 min at 4°C then rinsed four times in 0.1 M cacodylate buffer, pH 7.2. Cells were then post-fixed with 1% osmium tetroxide in 0.1 M cacodylate buffer, pH 7.2 for 1 hr at 4°C followed by three rinses. For SEM, cells were then immersed in 1% tannic acid in 0.05 M cacodylate buffer for 1 hr at RT before rinsing into 0.05 M cacodylate buffer and then into water. Cells were then incubated with 0.5% uranyl acetate in MilliQ water for 1 hr at RT, rinsed into water, then stored overnight in 0.05 M cacodylate buffer at 4°C. Cells were then dehydrated using a series of increasing ethanol concentration washes until equilibrated with 100% anhydrous ethanol where the cells were then critical point dried with liquid $CO_2$. The samples were then mounted onto aluminum specimen mounts with a conductive carbon paint and dried overnight using desiccation. The desiccated cells were then sputter coated with gold/palladium in a polaron sputter coater (Model 5100) and stored under dissection conditions. Prepared cells were then imaged with a JSM-6060 scanning electron microscope from JEOL USA, Inc (Peabody, MA).

For TEM, post-fixed cells were dehydrated by immersion in stepwise ethanol baths starting at 35% ethanol and finishing at 100% ethanol with immersion for 10 min each. Cells were embedded in resin using a low-viscosity embedding kit (Electron Microscopy Sciences; Cat# 14300). Embedded samples were cut into 600-nm-thick sections using a Reichert Ultracut-II-Ultramicrotome with a diamond knife. Cut sections were collected on 3 mm copper 200-mesh grids. Grids were contrasted using 2% uranyl acetate for 6 min followed by lead citrate for 4 min. Images were collected on a Jeol 1400 transmission electron microscope at 80 kV.

## Confocal imaging

Fixed cell confocal imaging was done at RT (23°C) and was accomplished using a spinning-disk (Yokogawa CSU-X1; Intelligent Imaging Innovations) Leica DMi600B microscope with a spherical aberration correction device and either a ×100/1.46 NA or ×63/1.40 NA Leica objective. Images were captured using a Hamamatsu ORCA-Flash 4.0 camera metal-oxide semiconductor device, and Z-slices of acquired images were assembled using SlideBook 6 software (Intelligent Imaging Innovations). Maximum- or summed-intensity projections were then assembled in SlideBook 6 or ImageJ and exported to Adobe Illustrator for editing. Side projections were vertically expanded using Illustrator to increase visual clarity of apical/basal localization. ImageJ intensity profile built-in toolset was used for line scan quantifications. Super Resolution via Optical Re-assignment was performed using a Nikon Ti2 microscope configured with a Yokogawa CSU-W1 spinning-disk confocal unit and imaged using a Hamamatsu Quest camera run using Nikon NIS-Elements software.

## Live imaging

Live-cell imaging of cells expressing GFP-EBP50 (*Garbett and Bretscher, 2012*) was done using an inverted Leica DMi8 widefield microscope equipped with a Leica ×100 NA air objective, a Leica DFC 9000 GTC camera, Leica Application Suite X THUNDER deconvolution software, and Leica adaptive focus control. Jeg3 cells were transiently transfected 24 hr prior to imaging, and the cell medium was changed to FluoroBrite DMEM (Thermo Fisher Scientific; Cat# A1896701) with 10% FBS immediately before imaging. The cells were then kept in a 5% $CO_2$ humidified environmental chamber at 37°C throughout the duration of the imaging as cells were imaged every 1 min for 15 min. Deconvolution images of GFP-EBP50 were performed on the Leica DMi8 widefield microscope, Leica Application Suite X THUNDER deconvolution software, and Leica adaptive focus control listed above. The default small sample Leica THUNDER deconvolution settings were used for this, apart from adjusting structure size to 1000 nm and reducing deconvolution strength to 30%.

## Statistical methods

Statistical comparisons were performed in GraphPad Prism. The type of statistical tests utilized, as well as the number of independent data points (*n*) and the configuration of error bars, is detailed in the figure legends respective to the data tested. The choice of technical or biological replicas was decided on an experiment-by-experiment basis. Nonparametric or parametric testing was justified through the assumption of the tested data to have a normal distribution or not. Any additional analyses related to specific techniques are described in the associated specific 'Materials and methods' sections.

## Super-resolution SIM

Structured Illumination Microscopy was performed using a Zeiss Elyra super-resolution inverted Axio Observer.Z1 microscope provided by Cornell University's Institute of Biotechnology Imaging Facility. Illumination was performed using lasers emitting 405, 488, 561, and 640 nm wavelengths through a ×63/1.4 NA oil objective and capturing images on a pco.edge 5.5m camera. Exposure time ranged from 50 to 400 ms and was dependent on sample and channel to optimize SIM reconstruction. As calculated through the ZEN black software, the number of Z-slice steps was set to the optimized minimum for each illumination channel. SIM grating was set to five rotations for all conditions, and processing was completed using default settings from the automatic SIM processing toolset provided by the ZEN software. By applying the built-in color alignment, ZEN software to a Z-stack of images containing 100 nm multicolor beads from an Invitrogen Molecular Probes TetraSpeck Fluorescent Microspheres Size Kit (Cat# T14792), color channels were corrected for chromatic aberration.

## Protein structural prediction

Full-length human ARHGAP18 sequence (amino acid M1 to L673) and human ezrin FERM (amino acid P2 to K296) were used to predict the structure and protein–protein interactions using AlphaFold2 (*Jumper et al., 2021*) accessed through the COSMIC[2] platform (*Cianfrocco et al., 2017*). Predictions were then entered into Open-Source PyMOL for coloration and raytracing.

## Reagents and cDNA

Ezrin antibodies were a mouse anti-ezrin antibody (Developmental Studies Hybridoma Bank; Cat# CPTC-ezrin-1; Research Resource Identification AB_2100318) used at 1:1,250 (western blot) or 1:100 (immunofluorescence), or a previously characterized polyclonal rabbit antibody raised against full-length human ezrin (APB B64) used at 1:1,000 (western blot) or 1:200 (immunofluorescence) (*Zaman et al., 2021*). A previously characterized homemade rabbit anti-phospho-ERM antibody, raised against recombinant phosphopeptide, was used at 1:1000 (western blot) (*Zaman et al., 2021*). Quantification of phospho-ERM rescue in ARHGAP18$^{-/-}$ cells included GFP-AHGAP18 and ARHGAP18-flag rescue conditions. Flag antibodies were rabbit anti-Flag (Cell Signaling Technology; Cat# 14793) used at 1:500 (western blot) or 1:500 (immunofluorescence), and mouse anti-flag M2 (Sigma-Aldrich; Cat# F1804) which was used at 1:1000 (western blot) or 1:200 (immunofluorescence). Mouse anti-GFP (B-2) (Santa Cruz Biotechnology, Inc; Cat# SC-9996) was used at 1:1000 (western blot). Mouse anti-actin C4 (EMD Millipore; Cat# MAB1501) was used at 1:1000. Rabbit anti-non muscle myosin-2A (BioLegend; Cat# 909801), myosin-2B (BioLegend; Cat# 909902), or myosin-2C (Cell Signaling Technology; Cat#

8189S) was used at 1:100 (western blot and immunofluorescence). Anti-myosin light chain antibodies were rabbit anti-MLC2 (Cat# 3672) used at 1:500 (western blot) and 1:50 (immunofluorescence) rabbit anti-phospho-MLC2 raised against phospho-Thr18/Ser19 (Cell Signaling Technology; Cat# 3674) used at 1:500 (western blot) and 1:50 (immunofluorescence), mouse anti-phospho-MLC2 raised against phospho-Ser19 (Cell Signaling Technology; Cat#3675S) used at 1:50 (immunofluorescence). Anti-cofilin and anti-phospho-cofilin antibodies (Abcam; Cat# ab12866) were used at 1:1000 (western blot). Rabbit anti-RhoA (Cell Signaling Technology; Cat# 2117) was used at 1:500 (western blot). Mouse anti-tubulin (Sigma-Aldrich; Cat# T5168) was used at 1:5000 (western blot). To stain actin, Alexa Fluor anti-actin 647 phalloidin (Invitrogen; Cat# A30107) was used at 1:25 (immunofluorescence). For nuclear staining, DAPI (Invitrogen; Cat# D1306) was used at 1:250 (immunofluorescence). Secondary antibodies used for immunofluorescence were anti-mouse Alexa Fluor 488 (LI-COR Biosciences; Cat# A21202) and anti-rabbit Alexa Fluor 568 (LI-COR Biosciences; Cat# A11036), both used at 1:250. Caly-culin A was purchased from Enzo Life Sciences (Cat# BML-El92-0100). pEGFP-RhoA Biosensor was a gift from Michael Glotzer (Addgene plasmid # 68026; RRID:Addgene_68026).

Human ARHGAP18 constructs were created using polymerase chain reaction (PCR) using New England Biolabs (NEB) Phusion High-Fidelity PCR Kit (Cat# E0553L) off a construct originally derived from the Harvard Plasma Database (ID # HsCD00379004). PCR and Gibson Assembly products were purified using NEB Monarch PCR & DNA Cleanup Kit (Cat# T1030S) and Thermo Scientific GeneJET Gel Extraction Kit (Cat# K0692). These products were cloned into the mammalian expression vector PQCXIP using NEB Gibson Assembly cloning Kit (Cat# E5510S), then transformed into OneShot TOP10 bacteria (Thermo Fisher; Cat# C404010). Successfully cloned constructs were selected using ampicillin resistance and then sequenced for verification. Single-sight mutagenesis for the gap-dead construct was performed using New England Biolab Q5 Site-Directed Mutagenesis Kit (Cat# E0554S). EBP50 tail sequence was generated by PCR off the GFP-EBP50 construct described below and inserted into the GAP-flag-EBP50(tail) vector using Gibson assembly. GFP-tagged ARHGAP18 used as a control in antibody quality control was a generous gift from Jennifer Gamble at the University of Sydney. Trans-fections were performed using PEI MAX polyethylenimine reagent (Polysciences; Cat# 24765).

## ARHGAP18 antibody

Purified human ARHGAP18 was produced by bacterial expression of an N-terminal-SUMO-HIS-tagged protein purified using a NiNTA resin. The SUMO tag was cleaved using purified ULP1 and then the ARHGAP18 was further purified using a HiLoad 16/600 superdex 200 size-exclusion chromatography installed on a General Electric AKTA pure FPLC. For antibody production, this purified protein was exchanged into PBS and denatured by adding 1 mM DTT and then boiled for 5 min at a final concen-tration of 2.1 mg/mL before flash freezing in liquid nitrogen. The frozen ARHGAP18 antigen was then shipped to Pocono Rabbit Farm & Laboratory, Inc (Canadensis, PA) for antibody production in rabbits. ARHGAP18 antibody specificity was confirmed by western blot at a concentration of (1:1000) against the antigen, the ARHGAP18[-/-] cells, and observation of a band shift when expressing a GFP-tagged variant of ARHGAP18 in Jeg3 cells (*Figure 2C*). The animal use protocol approved by Cornell Univer-sity was IACUC number 2014-0109 to A. Bretscher.

## Isothermal calorimetry (ITC)

Purified ARHGAP18 was produced as described in the antibody production except that the SUMO tag was not cleaved as it was found that the protein was insoluble above 2 mg/mL without it. Purified ezrin and FERM were produced as described in *Viswanatha et al., 2012*. Ezrin or FERM were diluted to 75–80 μM depending on the purified volume and number of injections needed for ITC. The proteins were then injected into 30 μM SUMO-ARHGAP18 using automated injections on a TA Instruments-Waters LLC (New Castle, DE) Affinity ITC-LV. Injections were performed using ITC run software (TA Instruments) with a minimum pulse time of 200 s and 2–5 μL injections and a stirring rate of 175 rotations per minute. The system was calibrated using the injection of water into water as a negative control and the injection of 0.95 mM $CaCl_2$ into 0.150 mM EDTA as a positive control. Injections were continued until saturation of the binding was observed through a plateau of the released energy per injection or the max volume of the injection syringe was depleted. Collected ITC data were then imported into NanoAnalyze software (TA Instruments), and a baseline linear curve representing the energy from injecting water into water was subtracted leaving a normalized energy injection dataset.

Injection points from either the first injection or injections after the binding affinity had plateaued were excluded and the remaining data points were then fit to a single independent binding curve. FL-ezrin or FERM injections that did not show an absorbance curve were not fit to one in the NanoAnalyze program as part of a pre-exclusion criteria.

## Force indentation

Cells were plated at 50% confluence to form uniform monolayers. Four hours prior to indentation, media was exchanged with fresh media containing 10 µM HEPES buffer. Cells were then mounted on an inverted epifluorescence microscope (Observer Z1, Zeiss), and force indentation curves were collected using a nanoindentation device (Chiaro, Optics11, Netherlands). Measurements were performed at RT with a 0.030 N/m cantilever with a spherical tip of 3 µm radius. Young's elastic modulus was calculated using the Optics11 DataViewer software using by applying a Hertzian model (equation below). Any indentation with a fit for the Hertzian model below an $R^2$ value of 0.8 was excluded from further analysis. Visualization of each indentation fit was performed by graphing the force as a function of displacement using the equation $F = \frac{4}{3} * E * \sqrt{PR} * h^{\frac{3}{2}}$, where $F$ is the force in Newton's, $E$ is the measured Young's modulus, $PR$ is the probe radius, and $h$ is displacement in meters starting at the point of contact. Graphing and analysis of Young's modulus were performed using GraphPad Prism.

## RhoA GAP assay

RhoA GAP assays were performed using a kit provided by Cytoskeleton Inc (Denver, CO; Cat# BK105) and by following the provided kit protocol summarized below. FERM ezrin and ARHGAP18 were purified as described above and were dialyzed into fresh GAP assay buffer provided in the kit. RhoA and p50 RhoGAP proteins were provided in the kit. For each reaction, ±4 µg of RhoA, 4 µg FERM, 4 µg ARHGAP18, or 6 µg p50RhoA GAP, depending on the experimental condition, was brought to a final concentration of 30 µL in GAP buffer. All reactions for the various conditions were prepared and run simultaneously in a single 96-well dish for each replicate. The reactions were started simultaneously by adding 200 µM GTP to the wells using a multichannel pipette. The reaction was incubated at 37°C for 20 min before adding CytoPhos reagent, which increased absorbance at 550 nm as free phosphate is released. The absorbance signal was measured at 550 nm using a Molecular Dynamics SpectraMax i3X, and an average normalized signal was calculated and then plotted in GraphPad Prism.

## Acknowledgements

We thank A Martin and N Buffard for their helpful discussions, M Graef for use of the SoRa microscope, J Lammerding, N Zuela-Sopilniak, and A Varlet for expertise with force indentation, and S Suzuki for pulldown advice. This work was supported by the National Institutes of Health grant R35GM131751 to A Bretscher and the Sam and Nancy Fleming Research Fellowship to A Lombardo. GFP-tagged ARHGAP18 used as a control in antibody quality control was a generous gift from J Gamble. Scanning electron microscopy was performed at the Microscopy Imaging Center at the University of Vermont (RRID# SCR_018821), and the Cornell Institute of Biotechnology Imaging Facility was supported by the National Science Foundation funding (1428922).

## Additional information

### Funding

| Funder | Grant reference number | Author |
|---|---|---|
| National Institute of General Medical Sciences | R35GM131751 | Andrew T Lombardo |
| The Sam and Nancy Fleming Research Fellowship | | Andrew T Lombardo |

| Funder | Grant reference number | Author |
|--------|------------------------|--------|

The funders had no role in study design, data collection and interpretation, or the decision to submit the work for publication.

## Author contributions

Andrew T Lombardo, Conceptualization, Data curation, Formal analysis, Funding acquisition, Validation, Investigation, Visualization, Methodology, Writing – original draft, Project administration, Writing – review and editing; Cameron AR Mitchell, Data curation, Formal analysis, Investigation, Writing – review and editing; Riasat Zaman, Resources, Validation, Investigation, Writing – review and editing; David J McDermitt, Resources, Investigation, Writing – review and editing; Anthony Bretscher, Conceptualization, Funding acquisition, Validation, Project administration, Writing – review and editing

## Author ORCIDs

Andrew T Lombardo ⬚ https://orcid.org/0000-0002-1814-5748
Anthony Bretscher ⬚ http://orcid.org/0000-0002-1122-8970

## Ethics

A custom ARHGAP18 antibody was produced in rabbit by Pocono Rabbit Farm & Laboratory, Inc (Canadensis, PA) using the animal use protocol approved by Cornell University IACUC number 2014-0109 to A. Bretscher.

## Decision letter and Author response

Decision letter https://doi.org/10.7554/eLife.83526.sa1
Author response https://doi.org/10.7554/eLife.83526.sa2

## Additional files

### Supplementary files

• MDAR checklist

• Supplementary file 1. Contains plasmids used, oligonucleotides used, and a table of all quantified data with summary statistics.

### Data availability

Supplemental document 1a provides all DNA plasmid information, 1b all DNA Oligos used and their sequence, and 1c all data used in statistical analysis and quantification. Full images of gels or blots shown are provided as supplements to the associated figure. All figures were assembled in Adobe Illustrator 2023 (Adobe, Mountain View, CA).

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
