## [Editor Report]

This important study demonstrates that ARHGAP18, through its recruitment and activation by ezrin, fine-tunes the local level of RhoA to allow for the appropriate distribution of actin-based structures between the microvilli and terminal web. The data use state-of-the-art imaging methods and strongly support the conclusions.

---

## [Decision Letter]

**Decision letter after peer review:**

Thank you for submitting your article "ARHGAP18-ezrin functions as an autoregulatory module for RhoA in the assembly of distinct actin-based structures" for consideration by *eLife*. Your article has been reviewed by 3 peer reviewers, one of whom is a member of our Board of Reviewing Editors, and the evaluation has been overseen by Anna Akhmanova as the Senior Editor.

Essential revisions:

1) Extend the time scale of the observations in the experiment shown in Figure 3 as requested by referee #1.

2) For the images in Figure 5C/D please also image with an antibody specific for the heavy chain of NM2B or, ideally conduct co-localization experiments with antibodies to both the phosphorylation RLC and to NM2B.

Comments, concerns and suggestions:

MV are more numerous and turnover faster in ARHGAP18-/- cells (Figure 3A,B). It seems that all of the MV in these cells are significantly shorter than those in control cells. Is that the case? Do they ever reach full height or is the turnover so fast that they never reach full length? Can this solely be accounted for by the increased NM2 activity or is it possible that dysregulation of the activity of an actin regulator(s), EPS8 or IRTKS for example, could account for this observed behavior?

While almost 100% of the WT microvilli do endure for the 10 min time period this would mean their average lifespan would be much longer, so extending the window of capture beyond 10 minutes would make the authors argument stronger, and also provide more insight into the dynamics of WT microvilli compared to the ARHGAP18-/- cells microvilli. Is there a technical reason the authors cannot image/track these structures for longer? Perhaps a different distribution plot instead of a bar graph would better show this data?

If RhoA is critical for terminal web and base of microvilli, then why don't we see any active RhoA in the middle of the wt cell? And are those punctate structures in the ARHGAP18 KO cell actually microvilli? A microvilli marker would be useful here.

There are several concerns about the imaging of myosin. The staining pattern for pRLC and NM2B in Figure 5C and D do not appear to be well co-localized. There was concern about the specificity of the PMLC antibody given prominent nuclear staining and that several other classes of myosin (M18, M15, M7a) are complexed with regulatory light chain. Thus, to strengthen this portion of the manuscript, the authors should also use the NM2B antibody in Figure 5D ,or better yet, co-label both myosin 2 and pRLC for the imaging in both Figure 5C and D. It was difficult to connect the myo2B signal to microvilli and the overlap between the imagining. Additional images that include microvilli labelling, with zoomed insets, would help solidify this observation. Did the authors try to image NM2A or NM2C? These are often expressed in epithelial cells? Can the authors comment on why NM2B was visualized in favor of these?

Apical 'contractile bundles' containing NM2B are missing in the ARHGAP18-/- cells while high levels of pMLC are seen across the apical region but are missing from cell junctions. Thus, is it hard to understand how contractile force is being generated to account for the increased stiffness given the distinct localizations of the motor and signal for an activated motor.

Cells lacking ARHGAP18-/- appear to make microvillar-like protrusions with a disrupted actin core (Figure 5E, F). Immunostaining for pMLC (Figure 5D) suggests to the authors that NM2 filaments have invaded microvilli. The morphology of the stained structures looks quite different from that of control cells and also from the small, dynamic MV shown in Figure 3A. Is it possible that the hyperactivated NM2 causes dysregulation of the actin cortex such that abnormal, non-MV protrusions are being made?

Figure 5F and G – It is difficult to discern what the authors were describing in Figure 5F and 5G in regards to actin density. The authors point to a single hole in the larger ARHGAP18 KO microvillus. However, we don't easily observe similar holes in the other microvilli in Figure 5G. Can the authors quantify this in some way?

Related, some zoomed insets for Figure 5E would be very helpful since it is difficult to actually see these images without using zoom features on the computer.

The loss of ARHGAP18 results in increased apical stiffness, as measured by AFM (Figure 5A), that is expected to be due to increased activity of NM2. However, the distribution of NM2B as shown by immunofluorescence seems quite different from that of pMLC (compare Figure 5C with 5D).

The affinity of the FERM domain for ARHGAP18 seems quite modest (appr 20 µM). While a detailed mapping of the interaction sites is not necessary, have the authors tested the longer ezrin that binds the GAP (1-479) the most (Figure 1C-E)?

Also, it is a bit puzzling to find that the phosphomimetic, that is presumably an 'open' state of ezrin with the FERM domain available for binding, appears to interact so weakly with ARHGAP18. While the interaction is certainly more significant that the full-length wild type ezrin it is quite a bit lower than the truncated mutants, which seems a bit unexpected. This may be because phosphomimetic replacements of Ser/Thr often do not actually mimic phosphorylation or only partially do so. This aspect of the data was glossed over and might deserve more explanation.

Specific comments:

Figure 1E – The legend states that WT and LOK/SLK KO conditions are significantly different. Is this for E? They compared all populations (Empty-flag through EzrinT567A flag)? Some clarification would help the reader here.

Line 139: They authors state " Cells expressing the construct had reduced numbers of microvilli depending on the expression level compared to wildtype Jeg3 cells (Figure 2A)." This is interesting. Can they quantify this? Do they have any insight into how much overexpression they are observing?

If I understand the methods, the authors are using a single clone of ARHGAP18 KO. While they importantly can rescue the phenotype with expression of ARHGAP18, have the authors observed similar phenotypes in other ARHGAP18 KO clones?

The Authors show that microvilli number and length is dependent on ARHGAP18-Ezrin regulation of RhoA but a discussion of the impact microvilli number and length has on cell and tissue function is missing and could help the reader appreciate the implications of this work.

Figure 1E, Figure 2D: How were the blots normalized for quantification?

Figure 2A, E: Where are the insets taken from in these cells?

Figure 4: Green on black is a little hard to see low level signals. For single channel imaging, the authors might consider using greyscale or better yet, inverted greyscale to enable better visualization of low signal above background.

Figure 4G and 4H: While this RhoA biosensor is indeed established, the aggregate nature of this signal is slightly concerning. In our experience with this biosensor, the lowest expressing cells are likely the most accurate for RhoA activity, while moderate to high expression leads to more aggregate/condensates forming. Were the authors able to observe any differences between low expressing cells and high expressing cells and what level of expression are shown in the figure. The biosensor from Goedhart might be useful as well here (https://pubmed.ncbi.nlm.nih.gov/34357388/).

Figure 5C: Use an arrow to indicate a contractile bundle.

Figure 5D: Explain the colors used in the merged and inset images.

Figure 5D: Discuss the nuclear staining with the pMLC antibody.

Figure 5E, left: It is quite difficult to discern the terminal web actin filaments indicated by the blue arrow. One can see them if they look at it for a while but they are not all that obvious at first look.

Line 222: This wording is awkward. The target of RhoA is to indirectly increase the phosphorylation levels of pMLC. Perhaps say "…, one results of RhoA activation is to increase the levels of pMLC phosphorylation…" Also, I prefer the authors discuss this in terms of myosin regulatory light chain (RLC) since there are two types of lights bound to this myosin.

Line 578: "as appositive" should be "as a positive".

Figure 6 (model) – Please indicate/label the red hexagon (presumably it is ARHGAP18) and explain what the two components of LOK/SLK (orange circle, grey triangle) are.

Video 1 is a bit too short and goes a bit too fast to really discern what is happening with the apical MV.

---

## [Author Response]

Essential revisions:1) Extend the time scale of the observations in the experiment shown in Figure 3 as requested by referee #1.

We have replaced Video 1 with a new supplemental video which doubles the time scale of the observation of EGFP-EBP50 labeled microvilli in live cells. A specific response to the reviewers’ comment can be found below.

2) For the images in Figure 5C/D please also image with an antibody specific for the heavy chain of NM2B or, ideally conduct co-localization experiments with antibodies to both the phosphorylation RLC and to NM2B.

We have added substantial new data regarding non-muscle myosin and the associated light chain phosphorylation. Main text Figure 5 now includes new SoRa super resolution confocal images of NM-myosin-2B localized to microvilli in ARHGAP18 deficient cells but absent from WT microvilli. Additionally, we have added a new supplemental figure (new Figure 5-supplemental figure 1) which includes confocal images of co-localized phospho-myosin light chain with non-muscle myosin2A, 2B and 2C in both WT and ARHGAP18 deficient cells. The original submission Figure S3 has been renamed to Figure 5-supplemental figure 2.

Comments, concerns and suggestions:MV are more numerous and turnover faster in ARHGAP18-/- cells (Figure 3A,B). It seems that all of the MV in these cells are significantly shorter than those in control cells. Is that the case? Do they ever reach full height or is the turnover so fast that they never reach full length? Can this solely be accounted for by the increased NM2 activity or is it possible that dysregulation of the activity of an actin regulator(s), EPS8 or IRTKS for example, could account for this observed behavior?

Addressing the question of microvilli length using fluorescence microscopy presented a number of technical hurdles. In response to suggestions by the senior editor, prior to reviews, we added SEM imaging specifically to characterize the shorter MV found in ARHGAP18 deficient cells. From the assembly of SEM images, it is clear that a great majority of MV are shorter and more variable in length in the ARHGAP18^-/-^ condition. However, we observe a small fraction of microvilli that achieve comparable lengths to microvilli on wild type cells. Our live cell tacking of GFP-EBP50 labeled microvilli, SEM and TEM imaging data all indicate that it is possible for the microvilli to achieve full height in ARHGAP18^-/-^ cells. However, the balance between turnover and maintenance is altered reducing the likelihood of achieving full length in ARHGAP18^-/-^ cells.

We have added additional text to the discussion to clarify this point.

Our GFP-EBP50 labeled microvilli indicates that biogenesis of new microvilli is increased in the ARHGAP18^-/-^ condition. We have ascribed this finding to the increased ERM activation and LOK/SLK activity found in ARHGAP18^-/-^ cells as part of our proposed feedback loop model. Recent data from other groups suggests that microvilli biogenesis involves at a minimum the combined efforts of EPS8, IRTKS and ERMs working in concert. Our data supports the broad outline of this proposed model of microvilli biogenesis, and we now acknowledge these other studies.

To address the reviewers’ comments, we have now included the following text in the discussion:

“Our live cell tracking of GFP-EBP50 labeled microvilli, SEM and TEM imaging data all indicate that it is possible for the microvilli in cells lacking ARHGAP18 to achieve full length. However, the balance between turnover and maintenance is altered reducing the likelihood of achieving full length.”

And

“Recent data suggests that microvilli biogenesis involves at a minimum the combined efforts of EPS8, IRTKS and ERMs working in concert and future work will be required to define how ARHGAP18 may interact with the activities of EPS8 or IRTKS (Gaeta et al., 2021).”

While almost 100% of the WT microvilli do endure for the 10 min time period this would mean their average lifespan would be much longer, so extending the window of capture beyond 10 minutes would make the authors argument stronger, and also provide more insight into the dynamics of WT microvilli compared to the ARHGAP18-/- cells microvilli. Is there a technical reason the authors cannot image/track these structures for longer? Perhaps a different distribution plot instead of a bar graph would better show this data?

The reviewer brings up an important oversight we made in writing the manuscript. We failed to mention that our group has reported the dynamics of microvilli on wild type Jeg3 cells using the same GFP-EBP50 marker and a separate ezrin-GFP marker that showed that “microvilli had lifetimes in the 7–15-minute range” (Garbett and Bretscher, 2012). This detailed characterization in WT jeg3 cells is the basis for the 10-minute timescale for our analysis. In this context, the finding that ARHGAP18 deficient cells’ microvilli have an average lifespan (5.6 ± 3.2 minutes) is significant. We have altered the text to correct this oversight. Additionally, we now provide a longer time series of both WT and ARHGAP18 KO cells. Thus, we have addressed the reviewers’ concerns with the following specific improvements:

1. We have replaced Video 1 showing EBP50-GFP in WT vs. ARHGAP18 KO cells with a time series that lasts 20 minutes. This new data is named Video 1 in the resubmission. This is double the length of the original Video (now renamed Video 2).

2. We have added the following text to the Results section:

“To see what effect the enhanced level of phospho-ezrin has on microvillar dynamics, we transfected wildtype and ARHGAP18^-/-^ cells with GFP-EBP50, a construct we have used for its excellent live cell imaging of microvilli, where we reported an average turnover rate of microvilli on the 7-15 minute timescale in WT Jeg-3 cells (Garbett and Bretscher, 2012).”

If RhoA is critical for terminal web and base of microvilli, then why don't we see any active RhoA in the middle of the wt cell? And are those punctate structures in the ARHGAP18 KO cell actually microvilli? A microvilli marker would be useful here.

The reviewers’ questions can be clarified with an explanation of the contributing factors and limitations of the cell lines and biosensor used in this study. Our images of WT cells are similar to previously published images from leaders in the field (see figure 1A in Priya et al., 2015).

We acknowledge that the signal in the center of the cell is not easily visible by the eye in the representative image of WT cells. The intensity signals plotted in Figure 4H are normalized to background and are thus all the trace is above background. While small in comparison to the junctional intensity, active RhoA signal is detectable by our quantification at the pixel level in the middle of WT cells and shown in Figure 4H. Because RhoA is at the top of a signaling cascade, we expect very small quantities of active RhoA to be biochemically effective in living cells.

We made significant effort to co-label the RhoA biosensor with a microvilli marker without success. The RhoA biosensor requires a specialized fixation method to avoid aggregation (see below reviewer comments) (see methods). Unfortunately, when we attempted to co-stain the RhoA biosensor with a microvilli marker, the markers did not work under these fixation conditions. Numerous publications using this RhoA biomarker show the marker alone as a single channel in fixed samples, likely for this reason (Priya et al., 2015, Liang et al., 2017, Vassilevet al., 2017).

There are several concerns about the imaging of myosin. The staining pattern for pRLC and NM2B in Figure 5C and D do not appear to be well co-localized. There was concern about the specificity of the PMLC antibody given prominent nuclear staining and that several other classes of myosin (M18, M15, M7a) are complexed with regulatory light chain. Thus, to strengthen this portion of the manuscript, the authors should also use the NM2B antibody in Figure 5D ,or better yet, co-label both myosin 2 and pRLC for the imaging in both Figure 5C and D. It was difficult to connect the myo2B signal to microvilli and the overlap between the imagining. Additional images that include microvilli labelling, with zoomed insets, would help solidify this observation. Did the authors try to image NM2A or NM2C? These are often expressed in epithelial cells? Can the authors comment on why NM2B was visualized in favor of these?

We appreciate the reviewers’ insight and have added new data that we feel greatly strengthens this manuscript. We put significant effort into imaging NM2B at the apical surface of our cells yet were prevented by several apparent technical hurdles from obtaining images of NM2B inside microvilli in ARHGAP18 deficient cells. In the time since the original submission, we have obtained access to a Super Resolution via Optical Re-assignment (SoRA) confocal within our department (see changes to Acknowledgements). The enhanced resolution combined with the benefits of confocal imaging has allowed us to overcome many of the technical hurdles preventing our confident imaging of NM2B inside microvilli. As suggested by the reviewer, we now include a new figure 5 subpanel (Figure 5D) showing a zoomed in inset of NM2B colocalized with the specific microvilli marker ezrin.

Additionally, we have added an new supplemental figure (Figure 5-supplemental figure 1) with confocal images of NM2A, NM2B, and NM2C colocalized with pMLC in both WT and ARHGAP18 KO cells. In our hands, the NM2B antibody stains better than NM2A antibody, producing clearer localization images, and allowing for the increased signal to noise required for the super resolution imaging techniques utilized within the manuscript. This benefit is the reason for its preferred use in our main text figures. We have added new text to the following sections regarding the NM-myo isoforms.

Results:

“We performed immunofluorescent confocal imaging of pMLC colocalized with each of the three non-muscle myosin-2 isoforms A,B, or C (Figure 5-supplemental figure 1). The side-by-side comparative localization of these non-muscle myosin-2 isoforms within WT Jeg-3 cells alone represents, to our knowledge, a first for human placental cells. In agreement with previously presented data by our group and others (Chinowsky et al. 2020, Zaman et al. 2021), isoforms 2A and 2B appear to localize to nearly identical structures including large contractile bundles and cell-cell junctions. Nonmuscle myosin-2C does not readily localize with stress fibers or larger contractile bundles in agreement with recent characterizations (Chinowsky et al. 2020). We then compared the localizations of non-muscle myosin-2 between WT and ARHGAP18 deficient cells (Figure 5C, Figure 5-supplemental figure 1).”

One reason the pMLC and NM2B colocalization was not included in the original manuscript is our concern over the quality of the mouse/rabbit antibody pairs required for this imaging. We have validated up to 6 different NM-myo antibodies (including a home-made one), yet all of these were produced in rabbit. The commercially produced mouse pMLC antibody required to do the colocalization, shows significant non-specific background in our hands. We have tested multiple commercially available pMLC antibodies, produced both in rabbit and in mouse and have the most confidence in the rabbit antibody shown in Figure 5E. The rabbit pMLC has some non-specific background staining to the nucleus but otherwise stains appropriate actin-based structures well. Therefore, we have chosen to present the localizations of NM2B or pMLC to microvilli separately because we have the most confidence in these immunofluorescent products (Figure 5D,E). None-the-less, based on the reviewer and editors’ suggestions we include the colocalization in the manuscript through supplemental Figure 5-supplemental figure 1 so the reader can see and assess these data.

The imaging of myosin through the entirety of a columnar cell inside of a 100-200nm microvillar rod is exceptionally difficult and feel we have made all reasonable efforts to satisfy the reviewers.

Apical 'contractile bundles' containing NM2B are missing in the ARHGAP18-/- cells while high levels of pMLC are seen across the apical region but are missing from cell junctions. Thus, is it hard to understand how contractile force is being generated to account for the increased stiffness given the distinct localizations of the motor and signal for an activated motor.

The exact question the reviewer is highlighting is characteristic of ARHGAP18 whose fly homologue is named conundrum “due to its unexpected phenotypes” (Neisch et al. 2013). The reviewer is correct that our data indicates that ARHGAP18 deficient cells have increased stiffness that appears to arise independent from NM2B contractile force. We believe that the increased apical stiffness comes from the freeing of actin monomers that would otherwise be sequestered into the longer microvilli in WT cells. These monomers are then incorporated into nearby actin networks at the apical surface, thus resulting in a stiffer apical region. We are not aware of any methods to directly test the hypothesis that additional free actin monomer from the roots of microvilli is being incorporated into structures that produce the increased stiffness, other than the provided TEM imaging. We believe the existing data supports our proposed model and that it represents the first potential mechanism to explain these unexpected phenotypes.

Additionally, the reviewer comments that pMLC appears to be missing from cell junctions in ARHGAP18^-/-^ cells. We have added new supplemental figures showing pMLC localizing at junctions in ARHGAP18 KO cells clarifying that the pMLC is not missing from cell junctions (Figure 5-supplemental figure 1).

We have added the following text to the discussion to clarify our understanding of the apical actin turnover and address the reviewers’ comment:

“It is possible that the increased turnover of microvilli core bundle actin results in the freeing of actin monomers that would normally be sequestered into the longer microvilli actin found in WT cells (Figure 6). The incorporation of these actin monomers into nearby structures may explain the increased apical stiffness measured in ARHGAP18 deficient cells (Figure 5 A,B).”

Cells lacking ARHGAP18-/- appear to make microvillar-like protrusions with a disrupted actin core (Figure 5E, F). Immunostaining for pMLC (Figure 5D) suggests to the authors that NM2 filaments have invaded microvilli. The morphology of the stained structures looks quite different from that of control cells and also from the small, dynamic MV shown in Figure 3A. Is it possible that the hyperactivated NM2 causes dysregulation of the actin cortex such that abnormal, non-MV protrusions are being made?

We have colocalized NM2 with the microvilli specific protein Ezrin in Figure 5D in response to the reviewers’ suggestion. These data rule out the possibility that the hyperactivated NM2 image is from non-MV protrusions.

Figure 5F and G – It is difficult to discern what the authors were describing in Figure 5F and 5G in regards to actin density. The authors point to a single hole in the larger ARHGAP18 KO microvillus. However, we don't easily observe similar holes in the other microvilli in Figure 5G. Can the authors quantify this in some way?Related, some zoomed insets for Figure 5E would be very helpful since it is difficult to actually see these images without using zoom features on the computer.

We edited Figures 5E-G to address the reviewers’ comments. We have removed the images in the original figure 5G as they are (1) redundant with Figure 5F and (2) are smaller. To help the reader discern what we are describing we have introduced a new, zoomed in, subfigure showing the alterations in microvilli actin between WT and ARHGAP18 deficient cells (New Figure 5G).

The loss of ARHGAP18 results in increased apical stiffness, as measured by AFM (Figure 5A), that is expected to be due to increased activity of NM2. However, the distribution of NM2B as shown by immunofluorescence seems quite different from that of pMLC (compare Figure 5C with 5D).

See the above comment on apical stiffness and the creation of free actin monomer in ARHGAP18 KO cells. We have edited the Discussion section to address this comment.

The affinity of the FERM domain for ARHGAP18 seems quite modest (appr 20 µM). While a detailed mapping of the interaction sites is not necessary, have the authors tested the longer ezrin that binds the GAP (1-479) the most (Figure 1C-E)?

The 19µM affinity was measured in free solution with just the two purified proteins, however the microvilli are effectively a small, confined space enclosed on 3 sides. This spatial confinement would lead to a tighter effective affinity in the microvilli. Additionally, our data suggest that there are cofactors or binding partners affecting the affinity in cells. Specifically, RhoA appears to bind at least transiently in a 3-part complex (Figure 4C,D). Detection of this multi-protein complex is not feasible using our available ITC technique and is beyond the scope of this manuscript. For these reasons, we think that the 19µM affinity may be an underestimate of the effective biological affinity in living cells. Our proposed model does not require tenacious attachment of ARHGAP18 to ezrin.

Expression of some ezrin constructs such as the 1-479 and the T567D are lethal or extremely disruptive to common expression systems. The ITC technique requires high purity of sample for confidence in the resulting data. We have attempted these experiments and have been unable to reasonably produce the suggested constructs at the appropriate purity and concentrations to perform the ITC experiments.

Also, it is a bit puzzling to find that the phosphomimetic, that is presumably an 'open' state of ezrin with the FERM domain available for binding, appears to interact so weakly with ARHGAP18. While the interaction is certainly more significant that the full-length wild type ezrin it is quite a bit lower than the truncated mutants, which seems a bit unexpected. This may be because phosphomimetic replacements of Ser/Thr often do not actually mimic phosphorylation or only partially do so. This aspect of the data was glossed over and might deserve more explanation.

The reviewer is correct that the phosphomimetic T567D does not mimic the fully open state. in vitro and in vivo data show that T567 is only partially open (see Chambers and Bretscher, 2005; Viswanatha et al., 2013; Pelasayed et al., 2017; Zaman et al., 2021). T567 is the most significant site regulated by phosphorylation, but other factors, like PIP_2_ and ezrin-binding proteins such EBP50 likely contribute to its degree of opening once T567 is phosphorylated.

We have added further explanation of these previous findings as requested by the reviewer in the Results section.

“Previous studies have shown that the ezrin-T567 phosphomimetic construct does not achieve a fully open state (Viswanatha et al., 2013; Pelasayed et al., 2017; Zaman et al., 2021). This is reflected in the western blot’s lower binding signal compared to the ezrin (1-583) construct which is fully open.”

Specific comments:Figure 1E – The legend states that WT and LOK/SLK KO conditions are significantly different. Is this for E? They compared all populations (Empty-flag through EzrinT567A flag)? Some clarification would help the reader here.

We have clarified the statement with the following text in the Figure 1 legend indicating the exact statistical test and data source:

“Quantification of normalized band intensity from all experiments presented in C and D. Bars represents mean ± SEM; n=4. Aggregated WT and LOK^-/-^SLK^-/-^ conditions as populations were compared using a ratio paired t-test and found to be significantly different (p=0.0235).”

Line 139: They authors state " Cells expressing the construct had reduced numbers of microvilli depending on the expression level compared to wildtype Jeg3 cells (Figure 2A)." This is interesting. Can they quantify this? Do they have any insight into how much overexpression they are observing?

Expression level was estimated by GFP fluorescence intensity but not quantified. The statement is generic to our studies as it is common among microvilli specific proteins that massive over expression leads to depletion of all microvilli. As a quality control mechanism, we always avoid focusing on the small subset of cells with blazing hot GFP signal that have no remaining microvilli after transfection.

If I understand the methods, the authors are using a single clone of ARHGAP18 KO. While they importantly can rescue the phenotype with expression of ARHGAP18, have the authors observed similar phenotypes in other ARHGAP18 KO clones?

We created two distinct KO lineages using separate CRISPR/CAS9 target guides. One guide to the start of the coding region at CTAACAGCCTACCACCCCAG and one guide starting at Serine18 with sequence CGGTCTGGTCCTTGCCGCTG. We first identified the core phenotypes of the ARHGAP18 KO cells, numerous shortened microvilli, in screening experiments in both populations of KOs. After preliminary screening of these mixed population lineages, we created 96 distinct clonal lines of ARHGAP18 KO which were then screened for survival under puromycin selection, total depletion of ARHGAP18 expression by Western blotting, and for uniform phenotype by immunofluorescence. Two colonialized cell lines were then confirmed to be KO cell lines with identical phenotypes. One of these ARHGAP18 KO cell line was then used for the data of this manuscript.

We have altered the methods to describe these quality control steps.

The Authors show that microvilli number and length is dependent on ARHGAP18-Ezrin regulation of RhoA but a discussion of the impact microvilli number and length has on cell and tissue function is missing and could help the reader appreciate the implications of this work.

We have proposed a model potentially explaining a question that has existed in the field for over 20 years. Exploration of our proposed model at the tissue level should be addressed in future studies.

Figure 1E, Figure 2D: How were the blots normalized for quantification?

The following text has been added to the methods section under western blotting:

“Blots were imaged using a Bio-Rad ChemiDoc, and band intensities were determined with ImageJ’s gel analysis package by normalizing band intensity against either cell housekeeping proteins (e.g. tubulin) or the unphosphorylated total protein of interest detected in the input sample, then calculating relative intensities.”

Figure 2A, E: Where are the insets taken from in these cells?

We have added boxes showing the location of the inset images for Figure 2A, E

Figure 4: Green on black is a little hard to see low level signals. For single channel imaging, the authors might consider using greyscale or better yet, inverted greyscale to enable better visualization of low signal above background.

The representation of Figure 4G was purposefully chosen to match the presentation of the RhoA biosensor from an influential manuscript where this biosensor was characterized at junctions (Priya et al., 2015). While we agree with the reviewers point, we choose to keep the representation as is so that our images can be compared to those existing in the literature that many readers will be familiar with.

Figure 4G and 4H: While this RhoA biosensor is indeed established, the aggregate nature of this signal is slightly concerning. In our experience with this biosensor, the lowest expressing cells are likely the most accurate for RhoA activity, while moderate to high expression leads to more aggregate/condensates forming. Were the authors able to observe any differences between low expressing cells and high expressing cells and what level of expression are shown in the figure. The biosensor from Goedhart might be useful as well here (https://pubmed.ncbi.nlm.nih.gov/34357388/).

We have taken the appropriate steps needed for confidence in this biosensor and have implemented stringent precautions and controls for its appropriate and accurate usage. Upon initial submission to *eLife*, the senior editor made specific suggestions to the authors regarding the rhoA biosensor. The authors sought out expert advice to improve the accuracy of their RhoA biosensor data which included the helpful suggestions made by the reviewer. Following the advice of the senior editor and experts in our field, we have already replicated and replaced the images and data presented to incorporate the suggestions made by the reviewer.

Figure 5C: Use an arrow to indicate a contractile bundle.

We have added arrows to Figure 5C.

Figure 5D: Explain the colors used in the merged and inset images.

We have added color definitions to the Figure now labeled as 5E.

Figure 5D: Discuss the nuclear staining with the pMLC antibody.

This comment has been addressed in the above responses.

Figure 5E, left: It is quite difficult to discern the terminal web actin filaments indicated by the blue arrow. One can see them if they look at it for a while but they are not all that obvious at first look.

We have provided zoomed in inset images of the microvilli actin in a new subpanel Figure 5G. Discerning the terminal web actin is more easily done at distance as the individual filaments become even harder to see when zoomed in on.

Line 222: This wording is awkward. The target of RhoA is to indirectly increase the phosphorylation levels of pMLC. Perhaps say "…, one results of RhoA activation is to increase the levels of pMLC phosphorylation…" Also, I prefer the authors discuss this in terms of myosin regulatory light chain (RLC) since there are two types of lights bound to this myosin.

We have altered this sentence to the following text:

“One result of RhoA activation is to increase the levels of phosphorylated myosin regulatory light chain (pMLC). We measured an increase in pMLC in ARHGAP18^-/-^ cells (Figure 4E, F).”

Line 578: "as appositive" should be "as a positive".

We have corrected this error.

Figure 6 (model) – Please indicate/label the red hexagon (presumably it is ARHGAP18) and explain what the two components of LOK/SLK (orange circle, grey triangle) are.

We have corrected these errors.

Video 1 is a bit too short and goes a bit too fast to really discern what is happening with the apical MV.

This comment has been addressed in the above responses.